# Generative Modeling with Bayesian Sample Inference

## Abstract

We derive a novel generative model from iterative Gaussian posterior inference. By treating the generated sample as an unknown variable, we can formulate the sampling process in the language of Bayesian probability. Our model uses a sequence of prediction and posterior update steps to iteratively narrow down the unknown sample starting from a broad initial belief. In addition to a rigorous theoretical analysis, we establish a connection between our model and diffusion models and show that it includes Bayesian Flow Networks (BFNs) as a special case. In our experiments, we demonstrate that our model improves sample quality on ImageNet32 over both BFNs and the closely related Variational Diffusion Models, while achieving equivalent log-likelihoods on ImageNet32 and ImageNet64.

## 1 Introduction

The field of deep learning has produced a multitude of generative models over the years (Harshvardhan et al., 2020). Variational autoencoders, for example, learn the data distribution by compressing data into a lower-dimensional representation (Kingma & Welling, 2013). Normalizing flows learn to map between a prior and the data distribution via invertible transformations, enabling exact likelihood computation (Rezende & Mohamed, 2015). Generative adversarial networks generate samples by pitting two models against each other such that one proposes artificial data samples while the other tries to distinguish real and generated (Goodfellow et al., 2014). Recently, diffusion models (DMs) have become a cornerstone of generative modeling (Sohl-Dickstein et al., 2015; Ho et al., 2020). They define a multi-step forward process that gradually adds noise to the data, turning it into pure noise. Then, a model is trained to reverse this process, enabling the generation of new data samples by starting from noise and iteratively denoising.

In this work, we take a Bayesian viewpoint of sample generation to propose a new generative model. Imagine that a sample $\boldsymbol{x}$ from the data distribution $\mathrm{p}(\boldsymbol{x})$ is fixed but unknown to us; however, we can receive noisy measurements $\boldsymbol{y}_i \sim \mathcal{N}(\boldsymbol{x}, \alpha_i^{-1})$ of it. Then, we can infer the unknown $\boldsymbol{x}$ by combining the information in these measurements. To be more precise, we start with a broad belief $\mathrm{p}(\boldsymbol{x}) = \mathcal{N}(\boldsymbol{x} \mid \boldsymbol{\mu}_0, \lambda_0^{-1})$ about $\boldsymbol{x}$ in the form of a Normal distribution with low precision $\lambda$, i.e. high variance, that encompasses the entire data distribution. Then, we can take a first noisy measurement $\boldsymbol{y}_1$ and form a posterior belief $\mathrm{p}(\boldsymbol{x} \mid \boldsymbol{y}_1)$ about the sample, which will be a little more precise and a little more correct. Iterating this process allows us to refine our estimate $\mathrm{p}(\boldsymbol{x} \mid \boldsymbol{y}_1, \dots, \boldsymbol{y}_k)$ to any desired level of precision.

We transform this inference process into a generative model by introducing a prediction model $f_{\boldsymbol{\theta}}$ that estimates $\boldsymbol{x}$ from our current Gaussian belief about it. Since the true $\boldsymbol{x}$ is unknown at generation time, we substitute it with an estimate $\hat{\boldsymbol{x}} = f_{\boldsymbol{\theta}}(\boldsymbol{\mu}_i, \lambda_i)$ and sample $\boldsymbol{y}_{i+1} \sim \mathcal{N}(\hat{\boldsymbol{x}}, \alpha_{i+1}^{-1})$ instead. Maximizing an evidence lower bound (ELBO) for the likelihood that this simple process assigns to the training data, trains $f_{\boldsymbol{\theta}}$ to reconstruct true $\boldsymbol{x}$ from uncertain belief states $(\boldsymbol{\mu}_i, \lambda_i)$ about them. Consequently, the noisy measurements $\boldsymbol{y}_i$ of predicted samples $\hat{\boldsymbol{x}}$ become indistinguishable from those of real samples $\boldsymbol{x}$, and our generative process converges toward producing new samples from the data distribution.

Our key **contributions** can be summarized as follows.

- We present a new generative model based on iterative posterior inference from noisy predictions.
- We derive an ELBO to enable effective likelihood optimization and show how we can reduce the variance of the training loss with importance sampling.

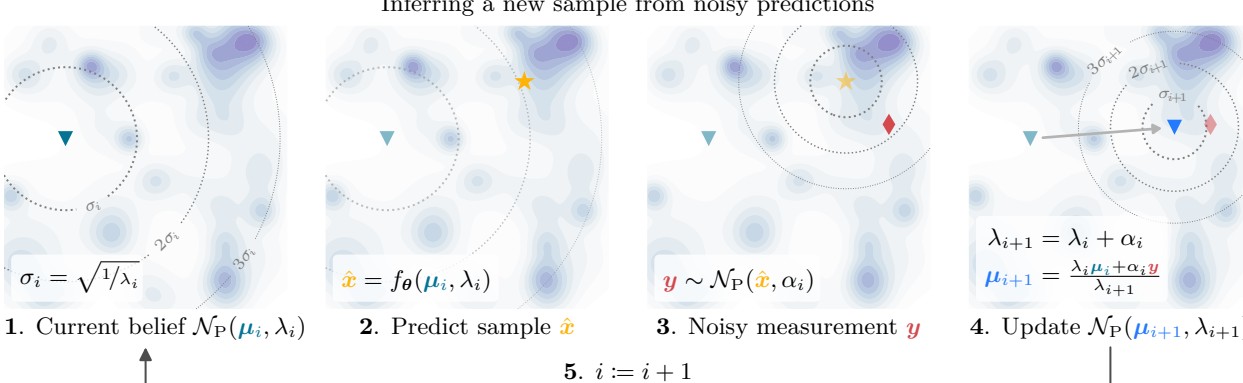

*Figure 1.* We view generation as the problem of inferring the identity of an unknown sample $\boldsymbol{x}$ from noisy observations. **1**. To begin, our belief about $\boldsymbol{x}$ is so broad as to cover the complete data distribution. **2**. We use a model $f_{\boldsymbol{\theta}}$ to guess which $\boldsymbol{x}$ likely corresponds to the information we have collected so far. **3**. Now, we pretend that $\hat{\boldsymbol{x}}$ is the true $\boldsymbol{x}$ and take a noisy measurement $\boldsymbol{y}$. **4**. We form the posterior belief about $\boldsymbol{x}$ to incorporate the information contained in $\boldsymbol{y}$. **5**. Repeat until we have identified a new sample with sufficient precision $\lambda_i$.

- Further, we compare our model in detail to Variational Diffusion Models (VDMs) (Kingma et al., 2023) and Bayesian Flow Networks (BFNs) (Graves et al., 2023).
- We show that the simple generative process described above includes BFN as a special case, providing a novel and simplified perspective on them, and analyze the relationship to DMs.
- Finally, we describe our model design and demonstrate empirically that our model surpasses both VDM and BFN in terms of sample quality on ImageNet32 while achieving equivalent log-likelihoods.

**Notation**   We parametrize Normal distributions either with a variance $\sigma^2$ as $\mathcal{N}(\boldsymbol{\mu}, \sigma^2 \boldsymbol{I})$ or with a precision $\lambda = 1/\sigma^2$ as $\mathcal{N}_{\mathrm{P}}(\boldsymbol{\mu}, \lambda \boldsymbol{I})$. Since all Normal distributions in this work are isotropic, we shorten these to $\mathcal{N}(\boldsymbol{\mu}, \sigma^2)$ and $\mathcal{N}_{\mathrm{P}}(\boldsymbol{\mu}, \lambda)$. $[n]$ is the set of integers $1, \dots, n$ and $\mathbb{R}_+$ refers to the non-negative reals.

## 2   Sample Discovery through Iterative Measurement

Consider a sample $\boldsymbol{x} \in \mathbb{R}^n$ that is unknown to us, but we can access noisy measurements $\boldsymbol{y}_i \sim \mathcal{N}_{\mathrm{P}}(\boldsymbol{x}, \alpha_i)$ of it. Then we can infer $\boldsymbol{x}$ from the sequence of measurements $\boldsymbol{y}_i$ through Bayesian inference. We start with a broad initial belief $\mathrm{p}(\boldsymbol{x}) \sim \mathcal{N}_{\mathrm{P}}(\boldsymbol{\mu}_0, \lambda_0)$ and update it with information contained in $\boldsymbol{y}_i$ per the following lemma.

**Lemma 2.1** (Posterior Update). *Let $\boldsymbol{x}, \boldsymbol{\mu} \in \mathbb{R}^n$ and $\lambda \in \mathbb{R}_+$ such that $\boldsymbol{x}$ is latent and $\mathrm{p}(\boldsymbol{x}) = \mathcal{N}_{\mathrm{P}}(\boldsymbol{x} \mid \boldsymbol{\mu}, \lambda)$ is a prior on $\boldsymbol{x}$; and $\boldsymbol{y} \sim \mathcal{N}_{\mathrm{P}}(\boldsymbol{x}, \alpha)$ where $\alpha \in \mathbb{R}_+$. Then the posterior is $\mathrm{p}(\boldsymbol{x} \mid \boldsymbol{y}) = \mathcal{N}_{\mathrm{P}}(\boldsymbol{x} \mid \boldsymbol{\mu}', \lambda')$ with*

$$\lambda' = \lambda + \alpha \quad and \quad \boldsymbol{\mu}' = 1/\lambda' \left[ \lambda \boldsymbol{\mu} + \alpha \boldsymbol{y} \right]. \tag{1}$$

*Proof.* See (Murphy, 2012, Section 4.4.1). $\qquad\square$

We can now iterate over the noisy measurements and update our belief until $\mathrm{p}(\boldsymbol{x} \mid \boldsymbol{y}_1, \dots, \boldsymbol{y}_k) \sim \mathcal{N}_{\mathrm{P}}(\boldsymbol{\mu}_k, \lambda_k)$ identifies $\boldsymbol{x}$ with sufficient precision. Sufficiency depends on the application but could be defined, for example in the case of images, such that most of the probability mass for each dimension of an image $\boldsymbol{x}$ is contained within a single color intensity bin of width $1/256$ for 8-bit color. Note that, at each step, all information contained in $\boldsymbol{y}_1, \dots, \boldsymbol{y}_k$ is captured in the current $\boldsymbol{\mu}_k$.

## 3   Sample Generation with Posterior Inference

We turn the procedure in Section 2 into a generative model, which we call *Bayesian Sample Inference* (BSI), as follows. We begin with an initial belief $(\boldsymbol{\mu}_0, \lambda_0)$ about the sample $\boldsymbol{x}$ which we will generate in the end,

with $\boldsymbol{\mu}_0$ sampled from a suitable prior distribution $\mathrm{p}(\boldsymbol{\mu}_0)$ and $\lambda_0$ fixed. Obviously, $\boldsymbol{x}$ is unknown a priori, so we cannot measure it, but we can estimate it from the information we have gathered so far.

Let $f_{\boldsymbol{\theta}} : \mathbb{R}^n \times \mathbb{R}_+ \to \mathbb{R}^n$ be a learned model with parameters $\boldsymbol{\theta}$ that estimates which unknown sample $\boldsymbol{x}$ we have observed so far from our current belief $(\boldsymbol{\mu}_i, \lambda_i)$. We estimate $\boldsymbol{x}$ as $\hat{\boldsymbol{x}}_{i-1} = f_{\boldsymbol{\theta}}(\boldsymbol{\mu}_{i-1}, \lambda_{i-1})$ and sample a noisy measurement $\boldsymbol{y}_i \sim \mathcal{N}_{\mathrm{P}}(\hat{\boldsymbol{x}}_{i-1}, \alpha_i)$ of $\hat{\boldsymbol{x}}_{i-1}$ in place of $\boldsymbol{x}$ with precision $\alpha_i$. Then, we can update our belief with $\boldsymbol{y}_i$ and Lemma 2.1 to the posterior $(\boldsymbol{\mu}_i, \lambda_i)$. Now, we alternate between these two steps, i.e. predicting and taking a noisy measurement followed by updating our current belief, until the posterior precision $\lambda_i$ is sufficient. Finally, we return $\hat{\boldsymbol{x}}^* = f_{\boldsymbol{\theta}}(\boldsymbol{\mu}_k, \lambda_k)$ as our generated sample. See Algorithm 1 for a formal description and Fig. 1 for a visual explanation.

*Algorithm 1.* Sampling with posterior inference
**input** Initial precision $\lambda_0$,
      precision schedule $\alpha_i$ for $i \in [k]$
**output** Sample $\hat{\boldsymbol{x}}^*$
1: Initialize belief $(\boldsymbol{\mu}_0, \lambda_0)$ with $\boldsymbol{\mu}_0 \sim \mathrm{p}(\boldsymbol{\mu}_0)$
2: **for** $i = 1, 2, \ldots, k$ **do**
3:    $\hat{\boldsymbol{x}}_{i-1} = f_{\boldsymbol{\theta}}(\boldsymbol{\mu}_{i-1}, \lambda_{i-1})$
4:    $\boldsymbol{y}_i \sim \mathcal{N}_{\mathrm{P}}(\hat{\boldsymbol{x}}_{i-1}, \alpha_i)$
5:    Update belief $\mathrm{p}(\boldsymbol{x} \mid \boldsymbol{y}_1, \ldots, \boldsymbol{y}_i) = \mathcal{N}_{\mathrm{P}}(\boldsymbol{\mu}_i, \lambda_i)$:
6:      $\lambda_i = \lambda_{i-1} + \alpha_i$
7:      $\boldsymbol{\mu}_i = {}^1\!/\lambda_i[\lambda_{i-1}\boldsymbol{\mu}_{i-1} + \alpha_i\boldsymbol{y}_i]$
8: **end for**
9: Return $\hat{\boldsymbol{x}}^* = f_{\boldsymbol{\theta}}(\boldsymbol{\mu}_k, \lambda_k)$

Since the posterior precision $\lambda_i$ does not depend on the generated sample $\hat{\boldsymbol{x}}_i$, we can choose the number of measurement rounds $k$ and precision schedule $\alpha_i$ a priori such that $\lambda_k$ will always be sufficiently large.

We have collected the proofs of all formal statements in this section in Appendix D.

## 3.1 Evidence Lower Bound

By interpreting BSI as a hierarchical latent variable model, we derive an ELBO (Kingma & Welling, 2013), i.e. a lower bound on $\log \mathrm{p}(\boldsymbol{x})$ assigned to a data point by our model. The ELBO will then serve as a natural training target for $f_{\boldsymbol{\theta}}$ to ensure that true data samples have high likelihood under our model.

We form our hierarchy out of the sequence of belief means $\{\boldsymbol{\mu}_i\}$, giving us

$$\mathrm{p}(\boldsymbol{x}) = \underset{\mathrm{p}(\boldsymbol{\mu}_0)\cdot\mathrm{p}(\boldsymbol{\mu}_1|\boldsymbol{\mu}_0)\cdots\mathrm{p}(\boldsymbol{\mu}_k|\boldsymbol{\mu}_{k-1}))}{\mathbb{E}}[\mathrm{p}(\boldsymbol{x} \mid \boldsymbol{\mu}_k)]. \tag{2}$$

The precisions $\{\lambda_i\}$ are not included as latent variables, because they do not depend on $\boldsymbol{x}$. With this hierarchy, we can derive the following ELBO.

**Theorem 3.1.** *Let $\boldsymbol{x} \in \mathbb{R}^n$ and $\alpha_{\mathrm{R}}, \alpha_i \in \mathbb{R}_+, i \in [k]$. Then the log-likelihood of $\boldsymbol{x}$ is lower-bounded as*

$$\log \mathrm{p}(\boldsymbol{x}) \geq -\mathcal{L}_{\mathrm{R}} - \mathcal{L}_{\mathrm{M}}^k \tag{3}$$

*by a reconstruction term $\mathcal{L}_{\mathrm{R}}$ and a measurement term $\mathcal{L}_{\mathrm{M}}^k$,*

$$\mathcal{L}_{\mathrm{R}} = \underset{\mathrm{q}(\boldsymbol{\mu}_k|\boldsymbol{x},\lambda_k)}{\mathbb{E}}\left[-\log \mathcal{N}_{\mathrm{P}}(\boldsymbol{x} \mid \hat{\boldsymbol{x}}_k, \alpha_{\mathrm{R}})\right] \quad and \quad \mathcal{L}_{\mathrm{M}}^k = \frac{k}{2}\underset{\substack{i\sim\mathcal{U}(0,k-1)\\\mathrm{q}(\boldsymbol{\mu}_i|\boldsymbol{x},\lambda_i)}}{\mathbb{E}}\left[\alpha_{i+1}\|\boldsymbol{x} - \hat{\boldsymbol{x}}_i\|_2^2\right] \tag{4}$$

*where*

$$\mathrm{q}(\boldsymbol{\mu}_i \mid \boldsymbol{x}, \lambda_i) = \underset{\mathrm{p}(\boldsymbol{\mu}_0)}{\mathbb{E}}\left[\mathrm{p}(\boldsymbol{\mu}_i \mid \boldsymbol{\mu}_0, \boldsymbol{x}, \lambda_i)\right], \quad \hat{\boldsymbol{x}}_i = f_{\boldsymbol{\theta}}(\boldsymbol{\mu}_i, \lambda_i) \quad and \quad \lambda_i = \lambda_0 + \sum_{j=1}^{i} \alpha_j. \tag{5}$$

The measurement term $\mathcal{L}_{\mathrm{M}}^k$ corresponds to the noisy measurement and update loop in Algorithm 1 and $\mathcal{L}_{\mathrm{R}}$ to the final computation of the sample $\hat{\boldsymbol{x}}^*$. $\mathrm{q}(\boldsymbol{\mu}_i \mid \boldsymbol{x}, \lambda_i)$ is the distribution of our belief $(\boldsymbol{\mu}_i, \lambda_i)$ about the unknown sample $\boldsymbol{x}$ after $i$ steps if we would have observed the true $\boldsymbol{x}$ instead of $\hat{\boldsymbol{x}}_1, \ldots, \hat{\boldsymbol{x}}_i$. $\mathrm{p}(\boldsymbol{\mu}_i \mid \boldsymbol{\mu}_0, \boldsymbol{x}, \lambda_i)$ is the marginal distribution of possible posterior beliefs $(\boldsymbol{\mu}_i, \lambda_i)$ with posterior precision $\lambda_i$ reachable from an initial belief $(\boldsymbol{\mu}_0, \lambda_0)$. Equivalently, $\mathrm{p}(\boldsymbol{\mu}_i \mid \boldsymbol{\mu}_0, \boldsymbol{x}, \lambda_i)$ is the distribution of beliefs $(\boldsymbol{\mu}_i, \lambda_i)$ after updating our initial belief $(\boldsymbol{\mu}_0, \lambda_0)$ with a single measurement of $\boldsymbol{x}$ with Lemma 2.1 – marginalized over all possible noisy measurements $\boldsymbol{y}$ at precision $\alpha = \lambda_i - \lambda_0$.

On closer examination, we see that $\mathcal{L}_{\mathrm{R}}$, measuring how accurately we can reconstruct $\boldsymbol{x}$ at the end, only depends on the total precision $\lambda_k$ that we accumulated in the first phase of the algorithm. However, $\mathcal{L}_{\mathrm{M}}^k$

depends both on the number of rounds $k$ and the precision schedule $\alpha_i$. We can derive an ELBO that is independent of $k$ and $\alpha_i$ by considering the limit as $k \to \infty$ and refining the precision schedule $\{\alpha_i\}_{i=1}^k$ into smaller and smaller steps while keeping the total precision $\alpha_M = \sum_{i=1}^k \alpha_i$ constant.

**Theorem 3.2.** *Let $\alpha_R, \alpha_M \in \mathbb{R}_+$. For any sequence of precision schedules $\alpha_{k,i}$ for $k \in \mathbb{N}, i \in [k]$ such that $\sum_{i=1}^k \alpha_{k,i} = \alpha_M$ and the sequence of functions $[k] \to \mathbb{R}_+ : i \mapsto \alpha_{k,i}$ converges uniformly to 0, we can take the limit of Theorem 3.1 as $k \to \infty$ to get*

$$\mathcal{L}_R = \mathop{\mathbb{E}}_{q(\boldsymbol{\mu}_{\lambda_M}|\boldsymbol{x},\lambda_M)} \left[ -\log \mathcal{N}_P(\boldsymbol{x} \mid \hat{\boldsymbol{x}}_{\lambda_M}, \alpha_R) \right] \quad and \quad \mathcal{L}_M^\infty = \frac{\alpha_M}{2} \mathop{\mathbb{E}}_{\substack{\lambda \sim \mathcal{U}(\lambda_0, \lambda_M) \\ q(\boldsymbol{\mu}_\lambda|\boldsymbol{x},\lambda)}} \left[ \|\boldsymbol{x} - \hat{\boldsymbol{x}}_\lambda\|_2^2 \right] \tag{6}$$

*where $q(\boldsymbol{\mu}_\lambda \mid \boldsymbol{x}, \lambda) = \mathbb{E}_{p(\boldsymbol{\mu}_0)} \left[ p(\boldsymbol{\mu}_\lambda \mid \boldsymbol{\mu}_0, \boldsymbol{x}, \lambda) \right]$, $\lambda_M = \lambda_0 + \alpha_M$ and $\hat{\boldsymbol{x}}_\lambda = f_{\boldsymbol{\theta}}(\boldsymbol{\mu}_\lambda, \lambda)$.*

As long as our model is more accurate in reconstructing $\boldsymbol{x}$ from more precise measurements, a reasonable assumption, Theorem 3.2 is a tighter bound on the log-likelihood than Theorem 3.1. To see this, we rewrite $\mathcal{L}_M^\infty$ in terms of the expected squared error at belief precision $\lambda$

$$h(\lambda) = \mathop{\mathbb{E}}_{q(\boldsymbol{\mu}_\lambda|\boldsymbol{x},\lambda)} \|\boldsymbol{x} - \hat{\boldsymbol{x}}_\lambda\|_2^2 \tag{7}$$

as

$$\mathcal{L}_M^\infty = \frac{\alpha_M}{2} \mathop{\mathbb{E}}_{\lambda \sim \mathcal{U}(\lambda_0, \lambda_M)} [h(\lambda)] \tag{8}$$

for which we have the following result.

**Lemma 3.3.** *If $h$ is strictly decreasing, $\mathcal{L}_M^\infty < \mathcal{L}_M^k$ for any $k$ and any precision schedule $\{\alpha_i\}$.*

## 3.2 Prior Distribution

Let's consider possible priors of the form $p(\boldsymbol{\mu}_0) = \mathcal{N}_P(\boldsymbol{0}, \gamma_0)$ for our initial belief. Then we have the following result for the encoding distribution $q(\boldsymbol{\mu}_\lambda \mid \boldsymbol{x}, \lambda)$ in Theorems 3.1 and 3.2.

**Lemma 3.4.** *Let $\lambda_0, \gamma_0 \in \mathbb{R}_+$, $p(\boldsymbol{\mu}_0) = \mathcal{N}_P(\boldsymbol{0}, \gamma_0)$ and $\lambda \geq \lambda_0$. Then*

$$q(\boldsymbol{\mu}_\lambda \mid \boldsymbol{x}, \lambda) = \mathcal{N}_P \left( \frac{\lambda - \lambda_0}{\lambda} \boldsymbol{x}, \frac{\lambda^2}{\lambda - \lambda_0 + \lambda_0^2/\gamma_0} \right). \tag{9}$$

How should we choose $\gamma_0$? We start the sampling process with initial precision, i.e. confidence, $\lambda_0$. If $\lambda_0$ was larger than $\gamma_0$, we would be unreasonably confident in our initial belief, since we know that $\boldsymbol{\mu}_0$ has more uncertainty than $\lambda_0$. From this, we deduce that the reasonable range for $\gamma_0$ is $[\lambda_0, \infty]$. At the same time, we want to avoid unwarranted assumptions by the prior, so we choose $\gamma_0 = \lambda_0$ for our model, which also gives us a particularly simple form for the encoding distribution.

**Corollary 3.5.** *Let $\lambda_0 \in \mathbb{R}_+$, $p(\boldsymbol{\mu}_0) \sim \mathcal{N}_P(\boldsymbol{0}, \lambda_0)$ and $\lambda \geq \lambda_0$. Then*

$$q(\boldsymbol{\mu}_\lambda \mid \boldsymbol{x}, \lambda) = \mathcal{N}_P \left( \frac{\lambda - \lambda_0}{\lambda} \boldsymbol{x}, \lambda \right). \tag{10}$$

## 3.3 Variance Reduction

The squared distance $\|\boldsymbol{x} - \hat{\boldsymbol{x}}_\lambda\|_2^2$ in $\mathcal{L}_M^\infty$ will necessarily vary significantly across the range of $\lambda$ with large values for small $\lambda$ where $q(\boldsymbol{\mu}_\lambda \mid \boldsymbol{x}, \lambda) \approx p(\boldsymbol{\mu}_0)$ and small values for large $\lambda$ when $\boldsymbol{\mu}_\lambda \approx \boldsymbol{x}$. We can reduce the variance of Monte Carlo (MC) estimates of $\mathcal{L}_M^\infty$ for ELBO evaluation or gradient computation in training with importance sampling with a suitable proposal distribution $p(\lambda)$.

**Corollary 3.6.** *Let $p(\lambda)$ be a probability distribution with support $[\lambda_0, \lambda_M]$. Then we have*

$$\mathcal{L}_M^\infty = \frac{1}{2} \mathop{\mathbb{E}}_{\substack{\lambda \sim p(\lambda) \\ q(\boldsymbol{\mu}_\lambda|\boldsymbol{x},\lambda)}} \left[ \frac{1}{p(\lambda)} \|\boldsymbol{x} - \hat{\boldsymbol{x}}_\lambda\|_2^2 \right]. \tag{11}$$

We can further rewrite $\mathcal{L}_\mathrm{M}^\infty$ as

$$\mathcal{L}_\mathrm{M}^\infty = \frac{1}{2} \mathop{\mathbb{E}}_{\lambda \sim \mathrm{p}(\lambda)} \left[ \frac{h(\lambda)}{\mathrm{p}(\lambda)} \right] \tag{12}$$

with $h$ as defined in Eq. (7). To minimize the variance of MC estimates of $\mathcal{L}_\mathrm{M}^\infty$, we want to bring $h(\lambda)/\mathrm{p}(\lambda)$ as close to a constant as possible. If it were actually constant, the variance of the MC estimate would be zero.

Let's begin by examining $h$ more closely. If we approximate $f_{\boldsymbol{\theta}}$ as $f_{\boldsymbol{\theta}}(\boldsymbol{\mu}, \lambda) = \boldsymbol{\mu}$ and assume that $\boldsymbol{x}$ is normalized to zero mean and unit variance, we get the closed form

$$\mathop{\mathbb{E}}_{\boldsymbol{x}}[h(\lambda)] \propto \frac{\lambda_0^2}{\lambda^2} + \frac{1}{\lambda}. \tag{13}$$

While $f_{\boldsymbol{\theta}}(\boldsymbol{\mu}, \lambda) = \boldsymbol{\mu}$ might seem a crude approximation at first, it is not too far off for large $\lambda$ where the model just needs to predict a small correction to its input.

Eq. (13) suggests that we should choose $\mathrm{p}(\lambda) \propto \lambda_0^2/\lambda^2 + 1/\lambda$ to minimize the variance of MC estimates. While evaluating $\mathrm{p}(\lambda)$ is simple enough, we would need to invert its cumulative distribution function (CDF) numerically to sample from it. Instead, we recognize that $1/\lambda$ dominates $\lambda_0^2/\lambda^2$ except for the smallest $\lambda$ and choose $\mathrm{p}(\lambda) \propto 1/\lambda$, i.e. a standard Log-Uniform$(\lambda_0, \lambda_\mathrm{M})$ distribution.

### 3.4 Training & Sampling

We train our model with the ELBO from Theorem 3.2 by optimizing $2\mathcal{L}_\mathrm{M}^\infty/n$. We do not optimize $\mathcal{L}_\mathrm{R}$ directly as its

*Algorithm 2.* Estimating the BSI training loss
**input** Data sample $\boldsymbol{x}$
**output** Monte Carlo estimate of $\mathcal{L}_\mathrm{M}^\infty$
1: Sample $t \sim \mathcal{U}(0,1)$, $\boldsymbol{\varepsilon} \sim \mathcal{N}(\boldsymbol{0}, \boldsymbol{I})$
2: $\lambda = \exp\big((\log \lambda_\mathrm{M} - \log \lambda_0) \cdot t + \log(\lambda_0)\big)$
3: $\boldsymbol{\mu}_\lambda = {}^{(\lambda - \lambda_0)}/_\lambda \, \boldsymbol{x} + \sqrt{{}^1/_\lambda} \, \boldsymbol{\varepsilon}$
4: Return $(\log \lambda_\mathrm{M} - \log \lambda_0) \, \lambda \cdot \| \boldsymbol{x} - f_{\boldsymbol{\theta}}(\boldsymbol{\mu}_\lambda, \lambda) \|_2^2$

*Algorithm 3.* Sampling with BSI
**input** Initial precision $\lambda_0$,
     precision schedule $\alpha_i$ for $i \in [k]$
**output** Sample $\hat{\boldsymbol{x}}^*$
1: Sample $\boldsymbol{\varepsilon}_i \sim \mathcal{N}(\boldsymbol{0}, \boldsymbol{I}), i = 0, \dots, k$
2: $\boldsymbol{\mu}_0 = \sqrt{{}^1/_{\lambda_0}} \, \boldsymbol{\varepsilon}_0$
3: **for** $i = 1, 2, \dots, k$ **do**
4:    $\hat{\boldsymbol{x}}_{i-1} = f_{\boldsymbol{\theta}}(\boldsymbol{\mu}_{i-1}, \lambda_{i-1})$
5:    $\lambda_i = \lambda_{i-1} + \alpha_i$
6:    $\boldsymbol{\mu}_i = \lambda_i^{-1}\Big(\lambda_{i-1}\boldsymbol{\mu}_{i-1} + \alpha_i\big(\hat{\boldsymbol{x}}_{i-1} + \sqrt{{}^1/_{\alpha_i}}\boldsymbol{\varepsilon}_i\big)\Big)$
7: **end for**
8: Return $\hat{\boldsymbol{x}}^* = f_{\boldsymbol{\theta}}(\boldsymbol{\mu}_k, \lambda_k)$

magnitude is negligible for sufficiently large $\alpha_\mathrm{M}$ and it is structurally similar to $\mathcal{L}_\mathrm{M}^\infty$, i.e. both amount to a squared distance. Algorithm 2 shows the resulting algorithm with our belief prior $\mathrm{p}(\boldsymbol{\mu}_0)$ and proposal distribution $\mathrm{p}(\lambda)$. Similarly, Algorithm 3 implements the abstract Algorithm 1 with our belief prior.

## 4 Discussion

We are aware of two generative models that are closely related to BSI, BFN (Graves et al., 2023) and VDM (Kingma et al., 2023). BFNs are generative models motivated from an information theory perspective with a sender and a receiver communicating about the sample. As we show in Appendix A.1, BFNs are a special case of our framework in Section 3 if we translate them to the probabilistic perspective. They correspond to choosing $\gamma_0 = \infty$ and $\lambda_0 = 1$, meaning that sampling always starts from the deterministic belief $(\boldsymbol{\mu}_0, \lambda_0) = (\boldsymbol{0}, 1)$. In contrast, BSI chooses $\gamma_0 = \lambda_0$, i.e. the noise in the initial belief corresponds to our confidence in it, and leaves $\lambda_0$ as a hyperparameter, which we investigate in Section 6. VDM are a type of DM that have shown excellent performance in likelihood-based modeling. They are similar to BSI insofar as they specify the distribution of latent variables directly rather than defining a Markovian noising process as classical DMs do.

All three models admit an ELBO of the form

$$-\log \mathrm{p}(\boldsymbol{x}) \le \mathcal{L}_\mathrm{R} + \frac{\bar{\omega} - \omega}{2} \mathop{\mathbb{E}}_{\substack{\omega \sim \mathcal{U}(\omega, \bar{\omega}) \\ \mathrm{q}(\boldsymbol{\psi}_\omega | \boldsymbol{x}, \omega)}} \big[ \| \boldsymbol{x} - \hat{\boldsymbol{x}}_\omega \|_2^2 \big] \tag{14}$$

for a set of latent variables $\boldsymbol{\psi}$ at precision levels $\omega$ between $\omega$ and $\bar{\omega}$. For BSI and BFN, the precision level $\omega$ is the belief precision $\lambda$ between $\lambda_0$ and $\lambda_\mathrm{M}$ and $\boldsymbol{\psi}_\omega = \boldsymbol{\mu}_\lambda$. For VDM, the latent variables $\boldsymbol{\psi}$ are called $\boldsymbol{z}$ and they parametrize $\omega$ as the SNR $\nu$ between $e^{-5}$ and $e^{13.3}$.

Despite this shared ELBO form, the models vary significantly. Table 1 lists the encoding distribution $\mathrm{q}(\boldsymbol{\psi} \mid \boldsymbol{x}, \omega)$ for each model, their prior, from which they begin the sampling process, and the update step that

*Table 1.* Central structures of VDM, BFN and BSI. To improve comparability, we parametrize VDM in terms of the signal-to-noise ratio (SNR) $\nu$. BFN and BSI are parametrized with the belief precision $\lambda$ as introduced in Section 3. $\boldsymbol{\varepsilon}_i \sim \mathcal{N}(\boldsymbol{0}, \boldsymbol{I})$ is sampling noise.

| Model | ELBO Encoder q$(\boldsymbol{\psi} \mid \boldsymbol{x}, \omega)$ | Latent Prior | Update Step for Sampling |
|---|---|---|---|
| VDM | $q(\boldsymbol{z} \mid \boldsymbol{x}, \nu) = \mathcal{N}_{\mathrm{P}}\left(\sqrt{\frac{\nu}{1+\nu}}\,\boldsymbol{x}, 1+\nu\right)$ | $\boldsymbol{z}_T \sim \mathcal{N}_{\mathrm{P}}(\boldsymbol{0}, 1)$ | $\boldsymbol{z}_i = \dfrac{\sqrt{\nu_{i+1}(1+\nu_{i+1})}\,\boldsymbol{z}_{i+1} + (\nu_i - \nu_{i+1})\left(\hat{\boldsymbol{x}}_i + \sqrt{\frac{1}{\nu_i - \nu_{i+1}}}\,\boldsymbol{\varepsilon}_i\right)}{\sqrt{\nu_i(1+\nu_i)}}$ |
| BFN | $q(\boldsymbol{\mu} \mid \boldsymbol{x}, \lambda) = \mathcal{N}_{\mathrm{P}}((\lambda-1)/\lambda\,\boldsymbol{x}, \lambda^2/(\lambda-1))$ | $\boldsymbol{\mu}_0 = \boldsymbol{0}$ | $\boldsymbol{\mu}_i = \dfrac{\lambda_{i-1}\boldsymbol{\mu}_{i-1} + \alpha_i\left(\hat{\boldsymbol{x}}_{i-1} + \sqrt{\frac{1}{\alpha_i}}\,\boldsymbol{\varepsilon}_i\right)}{\lambda_{i-1} + \alpha_i}$ |
| BSI | $q(\boldsymbol{\mu} \mid \boldsymbol{x}, \lambda) = \mathcal{N}_{\mathrm{P}}((\lambda-\lambda_0)/\lambda\,\boldsymbol{x}, \lambda)$ | $\boldsymbol{\mu}_0 \sim \mathcal{N}_{\mathrm{P}}(\boldsymbol{0}, \lambda_0)$ | |

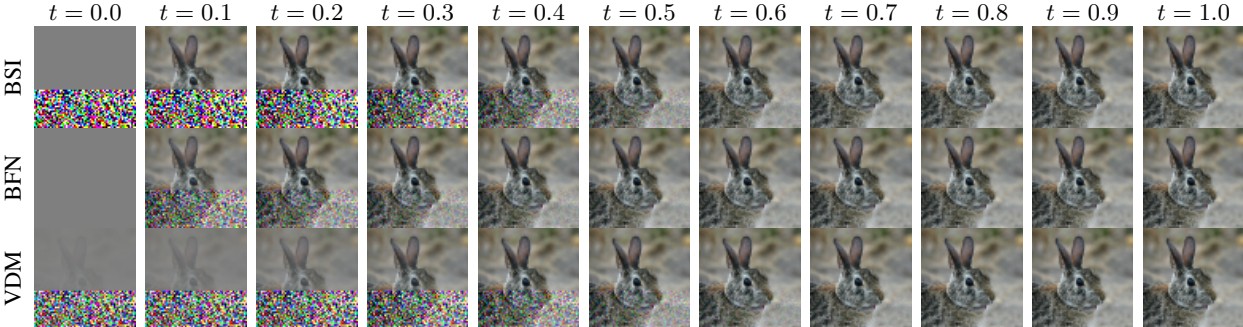

*Figure 2.* ELBO encoders q, i.e. training inputs, of BSI, BFN and VDM. $t$ parametrizes the precision levels by the respective model's precision schedule with $t = 0$ being pure noise, ideally, and $t = 1$ almost equaling the data. Top half shows the mean of q and bottom half a sample. Mean $\boldsymbol{0}$ is gray because all models rescale the data to $[-1, 1]$. BFNs apply little noise overall and reach a deterministic state at $t = 0$. For VDM, significant information about the sample is preserved in the structure of the mean at the highest noise level. In contrast, BSI converges to its latent prior distribution.

the models iterate during sampling. First, we see that VDM starts sampling from a standard Normal vector and BFN from the deterministic $\boldsymbol{0}$. Only BSI allows sampling from an initial precision $\lambda_0$ less than 1, which has been shown to improve sample diversity in consistency models (Song & Dhariwal, 2024). Second, the update step shared between BSI and BFN is significantly simpler than the VDM update with respect to the precision parameter and does not require evaluation in log-space for numerical stability as recommended for VDM (Kingma et al., 2023).

For the encoding distribution q$(\boldsymbol{\psi} \mid \boldsymbol{x}, \omega)$, which provides the training inputs when the models optimize their ELBO, we turn to Fig. 2. First, we note that BFN adds little noise overall due to their noise variance $(\lambda-1)/\lambda^2$ going to 0 for both small and large $\lambda$. Next, we notice the encoding distribution q$(\boldsymbol{\psi} \mid \boldsymbol{x}, \underline{\omega})$ with the most noise at $t = 0$. While it agrees exactly with the latent prior used for sampling for BSI and BFN, for VDM it becomes approximately $\mathcal{N}_{\mathrm{P}}(0.08\boldsymbol{x}, 1)$, which differs significantly from the standard Normal prior for sampling. In fact, the image motif is still clearly discernible in the distribution mean for VDM at its maximum noise level. The amount of signal remaining in the mean for BSI at high noise levels is counteracted by much higher noise variance, e.g. 15.85 at $t = 0.1$ for BSI compared to 0.96 for VDM.

**Diffusion Models**  If we currently hold the belief $(\boldsymbol{\mu}', \lambda')$, the distribution over beliefs $(\boldsymbol{\mu}, \lambda' - \alpha)$ that are $\alpha$ less precise is

$$p(\boldsymbol{\mu} \mid \boldsymbol{\mu}', \boldsymbol{x}) = \mathcal{N}\left(\xi^{-1}\left[\frac{\lambda\lambda'}{\alpha}\boldsymbol{\mu}' - \lambda_0\boldsymbol{x}\right], \xi\right) \qquad (15)$$

for a certain precision $\xi$. This shows that BSI can be written as a DM with a non-Markovian forward or "noising" process. See Appendix A.2 for a detailed derivation of this connection. There we also exploit that BFNs are a special case of BSI to derive the forward process for BFN and show that it is Markov, in contrast to the BSI process.

# 5    Model Design

In this section, we introduce a design for the prediction model in BSI. We begin by deriving a preconditioning structure for $f_{\boldsymbol{\theta}}$, i.e. a type of model structure similar to noise prediction in DMs. Then, we describe how we bring $\lambda$ into a suitable range as an input for deep learning. Finally, we give our choice of the hyperparameters $\lambda_0$, $\alpha_{\mathrm{M}}$ and $\alpha_{\mathrm{R}}$ and report the model architectures we used as the backbone of $f_{\boldsymbol{\theta}}$.

## 5.1    Preconditioning

It has long been known in the context of DMs that training models to predict $\boldsymbol{x}$ directly from a noisy input can hinder learning and limit sample quality (Karras et al., 2022; Ho et al., 2020). For probabilistic modeling, it is especially important that the model prediction stays close to the true sample if the input is already at a low noise level to achieve high ELBOs. This can be seen, for example, in Corollary 3.6 where prediction errors for high-precision input beliefs with large $\lambda$ have a higher weight. Instead of predicting $\boldsymbol{x}$, DMs commonly either predict a variation of the noise in the model input (Ho et al., 2020; Song et al., 2021a) or an adaptive mixture of the noise and the true sample (Salimans & Ho, 2021). In the end, these approaches amount to adding a skip connection to the model with specific weights.

For BSI, we derive such a preconditioning structure with the adaptive-mixture approach from Karras et al. (2022). Let $f'_{\boldsymbol{\theta}}$ be our neural network. Then we define the preconditioned $f_{\boldsymbol{\theta}}$ as

$$f_{\boldsymbol{\theta}}(\boldsymbol{\mu}, \lambda) = c_{\mathrm{skip}}\boldsymbol{\mu} + c_{\mathrm{out}}f'_{\boldsymbol{\theta}}(c_{\mathrm{in}}\boldsymbol{\mu}, \lambda) \tag{16}$$

and find the parameters through the conditions proposed by Karras et al. (2022). $c_{\mathrm{in}}$ and $c_{\mathrm{out}}$ are chosen such that the input to $f'_{\boldsymbol{\theta}}$ and its training target have unit variance. $c_{\mathrm{skip}}$ is then chosen to minimize $c_{\mathrm{out}}$, which minimizes the influence of prediction errors and ensures that $f_{\boldsymbol{\theta}}$ retains most of the signal already contained in $\boldsymbol{\mu}$ at large precisions $\lambda$.

From these conditions, we derive

$$c_{\mathrm{skip}} = {}^{(\lambda-\lambda_0)}\!/_{\kappa}, \quad c_{\mathrm{out}} = \sqrt{{}^1\!/_{\kappa}}, \quad c_{\mathrm{in}} = \sqrt{{}^{\lambda}\!/_{\kappa}} \tag{17}$$

where $\kappa = 1 + {}^{(\lambda-\lambda_0)^2}\!/_{\lambda}$ in Appendix C. $\lambda$ is the precision of our current belief about $\boldsymbol{x}$ and the input to $f_{\boldsymbol{\theta}}$.

## 5.2    Precision Encoding

The magnitude of $\lambda$ makes it impractical as a feature for neural networks. However, the CDF $F$ of $\mathrm{p}(\lambda)$ is a natural way to scale $\lambda$ from $[\lambda_0, \lambda_{\mathrm{M}}]$ to $[0, 1]$ as in DMs and flow matching (FM) (Lipman et al., 2023). In practice, we use $f_{\boldsymbol{\theta}}(\boldsymbol{\mu}, t)$ instead of $f_{\boldsymbol{\theta}}(\boldsymbol{\mu}, \lambda)$ where

$$t = F(\lambda) = \frac{\log \lambda - \log \lambda_0}{\log(\lambda_{\mathrm{M}}) - \log \lambda_0}. \tag{18}$$

Compared to linear re-scaling, our method makes it easier for $f_{\boldsymbol{\theta}}$ to distinguish belief precisions in the high-noise regime.

## 5.3    Hyperparameters

Apart from $f_{\boldsymbol{\theta}}$, BSI has three hyperparameters, $\lambda_0$, $\alpha_{\mathrm{M}}$ and $\alpha_{\mathrm{R}}$. $\lambda_0$ should be small enough that the initial belief covers the whole data distribution. We have found experimentally that $\lambda_0 = 10^{-2}$ optimizes likelihoods and sample quality at the same time for images rescaled to $[-1, 1]$, see Section 6.3. This agrees with the finding of Song & Dhariwal (2024) that large initial noise scales improve sample diversity in consistency models.

$\alpha_{\mathrm{M}}$ should be large enough that a noisy measurement at precision $\alpha_{\mathrm{M}}$ identifies an $\boldsymbol{x}$, e.g. for images almost all probability mass of $\mathcal{N}_{\mathrm{P}}(\boldsymbol{x}, \alpha_{\mathrm{M}})$ should be contained within a single 8-bit color intensity bin. We choose $\alpha_{\mathrm{M}} = 10^6$, which Graves et al. (2023) also picked for BFN. While $\mathcal{L}_{\mathrm{M}}^{\infty}$ dwarfs $\mathcal{L}_{\mathrm{R}}$, $\alpha_{\mathrm{R}} = 2\alpha_{\mathrm{M}}$ gives a slight edge in likelihood, empirically, as also observed by Graves et al. (2023).

### 5.4 Architecture

After the preconditioning and mapping $\lambda$ to a $t \in [0, 1]$, there are two more steps to turn the inputs $\boldsymbol{\mu}$ and $t$ of $f_{\boldsymbol{\theta}}'$ into effective features for a neural network. Regarding $t$, we convert it into a 32-dimensional precision embedding with a sinusoidal position encoding (Vaswani et al., 2017).

The Fourier features proposed by Kingma et al. (2023) are an essential component to reach high likelihoods, because they help the model distinguish fine details at high likelihoods, i.e. for inputs that are already close to the data distribution. They are basically a sinusoidal embedding of every dimension of $\boldsymbol{\mu}$. In particular, we extend $\boldsymbol{\mu}$ to the vector

$$\left(\boldsymbol{\mu} \quad \sin(2^i \pi \boldsymbol{\mu}) \quad \cos(2^i \pi \boldsymbol{\mu})\right) \ i \in n_{\min}, \dots, n_{\max} \tag{19}$$

before passing it into the neural network. We choose $n_{\min} = 6$ and $n_{\max} = 8$, in effect increasing the dimensionality of the input to the neural network from $n$ to $7n$.

For the neural network itself, we use two architectures, U-Nets (Ronneberger et al., 2015) and Vision Transformers (ViTs) (Dosovitskiy et al., 2020). We use the U-Net configuration proposed by Kingma et al. (2023) which adapts the widely used configuration from (Ho et al., 2020) for likelihood estimation. Most notably, the (Kingma et al., 2023) version has no downsampling between layers of the U-Net, which lets them increase the number of U-Net levels to 32.

ViTs are a more recent architecture inspired by the success of transformers (Vaswani et al., 2017). They represent images as a set of patches with a 2D position embedding and process them with global attention, in contrast to convolutional architectures like the U-Net where communication happens primarily locally. We opt for the Diffusion Transformer (DiT) architecture (Peebles & Xie, 2023) which has been shown to improve sample quality over variants of the (Ho et al., 2020) U-Net model.

## 6 Experiments

We evaluate BSI on the ImageNet (Deng et al., 2009) dataset in terms of log-likelihood and sample quality and on CIFAR10 (Krizhevsky, 2009) in terms of log-likelihood. While BSI as a method is general and not specific to images, we chose image datasets, because they are established benchmarks in the probabilistic modeling literature. In our experiments, we compare against BFN (Graves et al., 2023) and VDM (Kingma et al., 2023). BFNs are a special case of our framework (see Section 3) and provides an important reference point for the effect of the non-deterministic hyper-prior $p(\boldsymbol{\mu}_0)$ in BSI. VDMs are a representative of the diffusion family of models specifically designed for probabilistic modeling that is structurally similar to BSI as we explained in Section 4.

In Appendix B, we describe how we compute the ELBO, which we derived in Section 3.1 for continuous $\boldsymbol{x}$, on discretized images with 8-bit color channels. Appendix E lists hyperparameters and training details and Appendix F shows some generated samples.

### 6.1 ImageNet

For this evaluation, we train a DiT (Peebles & Xie, 2023) in the BFN, VDM and BSI model, respectively, on the official 32×32 and 64×64 versions of ImageNet (Chrabaszcz et al., 2017). We train each model from three seeds and evaluate the log-likelihood of the test set in bits per dimension (BPD) and the sample quality in terms of Fréchet inception distance (FID) against the test set. For the log-likelihood, we evaluate each model's ELBO with 5 samples from the respective equivalent of $\mathcal{L}_{\mathrm{M}}^{\infty}$ and 2 samples from the respective equivalent of $\mathcal{L}_{\mathrm{R}}$. For the sample quality, we draw $50\,000$ *unconditional* samples from each model with 1024 steps and then compute the FID between the generated samples and the test set. On the 32×32 resolution images, we train the DiT-L-2 configuration for $2\,\mathrm{M}$ steps and on the 64×64 resolution data, we train the DiT-L-4 configuration for $1\,\mathrm{M}$ steps.

*Table 2.* Log-likelihood in BPD and sample quality (FID) against the test set on ImageNet. We compute standard deviations over 3 seeds.

| Model | BPD ↓ | FID ↓ |
|---|---|---|
| ImageNet32 (2 M train steps) | | |
| BFN | $3.448 \pm 0.005$ | $11.0 \pm 0.1$ |
| VDM | $3.452 \pm 0.006$ | $9.9 \pm 0.5$ |
| BSI | $3.448 \pm 0.006$ | $8.9 \pm 0.1$ |
| ImageNet64 (1 M train steps) | | |
| BFN | $3.222 \pm 0.006$ | $38.2 \pm 0.8$ |
| VDM | $3.228 \pm 0.006$ | $35.2 \pm 0.7$ |
| BSI | $3.218 \pm 0.004$ | $30.1 \pm 0.4$ |

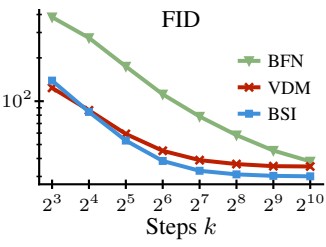 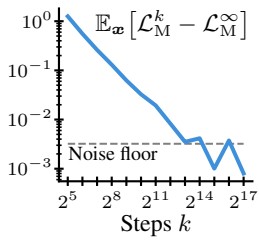 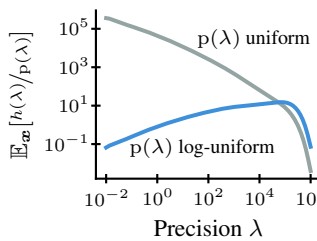 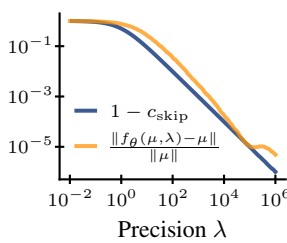

*Figure 3.* BSI's sample quality converges quickly and to a lower FID with increasing number of steps.

*Figure 4.* $\mathcal{L}_M^k$ converges to $\mathcal{L}_M^\infty$ from above as predicted in Lemma 3.3.

*Figure 5.* Our proposal distribution shrinks the range of $h(\lambda)/p(\lambda)$, reducing ELBO variance.

*Figure 6.* As $\lambda$ increases, $\hat{\boldsymbol{x}} = f_{\boldsymbol{\theta}}(\boldsymbol{\mu}, \lambda)$ and the belief $\boldsymbol{\mu}$ converge.

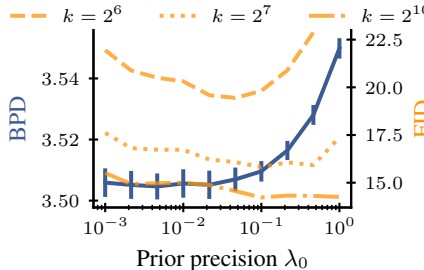 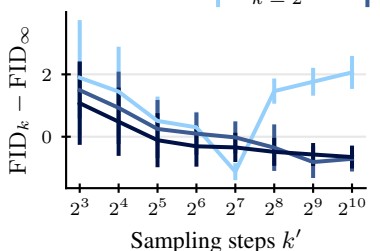 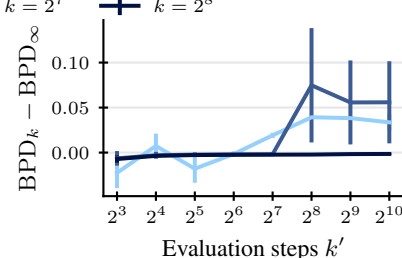

*Figure 7.* $\lambda_0$ balances likelihood and sample quality for varying sample steps $k$.

*Figure 8.* FID and likelihood difference between models trained on $\mathcal{L}_M^k$ and $\mathcal{L}_M^\infty$ when evaluated for $k'$ steps.

Table 2 shows that BSI achieves equivalent likelihoods to VDM and BFN while generating higher-quality samples in terms of FID. This aligns with the result for consistency models by Song & Dhariwal (2024) that a larger variance of the initial state – initial belief $\boldsymbol{\mu}_0$ for BSI – improves the sample diversity. Ordering the models by improving FID, we have BFN first with an initial variance of 0 ($\boldsymbol{\mu}_0 = \boldsymbol{0}$), then VDM with an initial variance of 1 and finally BSI with an initial variance of $\lambda_0^{-1} = 100$. The magnitude of the FID on ImageNet64 aligns with the results reported by Peebles & Xie (2023) after 1 M training steps. Furthermore, Fig. 3 shows that BSI achieves better sample quality than BFN and VDM for the same number of steps on ImageNet64.

**ELBO Convergence** Fig. 4 shows how the finite step ELBO from Theorem 3.1 converges towards its infinite step counterpart as $k \to \infty$ on the test set of ImageNet32. For this plot, we sampled 100 precisions $\lambda$ per image for the Monte Carlo estimates of $\mathcal{L}_M^k$ and $\mathcal{L}_M^\infty$. The convergence trend continues right to the noise floor where the noise overshadows the signal, marked in the plot by the standard deviation of the Monte Carlo estimator for the difference between the two terms.

## 6.2 CIFAR10

We train the same U-Net architecture as VDM (Kingma et al., 2023) and BFN (Graves et al., 2023) on CIFAR10. Table 4 shows that BSI achieves equivalent log-likelihoods in terms of BPD. Due to the significant number of training steps (10 M), we followed (Kingma et al., 2023; Graves et al., 2023) and trained only a single model on this dataset.

**Variance Reduction** Fig. 5 verifies the effect of importance sampling with a log-uniform distribution that we propose in Section 3.3. It reduces the range of the $h(\lambda)/p(\lambda)$ term in Eq. (12) by about 4 orders of magnitude on CIFAR10 and therefore the variance of a Monte Carlo estimate of the ELBO.

### 6.3 Parameter Studies

In the following, we evaluate the impact of our modeling and parameter choices. Unless otherwise stated, we trained each model for $100\,$k steps on ImageNet32 with a DiT architecture, evaluated the likelihood of the test set in BPD with the infinite-step ELBO and used 1024 sampling steps to compute the FID. We will verify the assumptions of the log-uniform proposal distribution $p(\lambda)$, compare DiT and U-Net model architectures, and evaluate the prior precision $\lambda_0$ and training on the finite-step ELBO $\mathcal{L}_M^k$.

**Proposal Distribution**  In Section 3.3, we have chosen a log-uniform proposal distribution $p(\lambda)$ based on the assumption that $f_{\boldsymbol{\theta}}(\boldsymbol{\mu}, \lambda) \approx \boldsymbol{\mu}$. Fig. 6 shows that the relative distance between $\boldsymbol{\mu}$ and $f_{\boldsymbol{\theta}}(\boldsymbol{\mu}, \lambda)$ falls quickly for $\lambda > 1$, when the belief $(\boldsymbol{\mu}, \lambda)$ contains enough information that the model mostly refines the belief. Our preconditioning structure $f_{\boldsymbol{\theta}}(\boldsymbol{\mu}, \lambda) = c_{\text{skip}}\boldsymbol{\mu} + c_{\text{out}}f'_{\boldsymbol{\theta}}(c_{\text{in}}\boldsymbol{\mu}, \lambda)$ derived in Section 5.1 ensures that $f_{\boldsymbol{\theta}}$ retains existing information as the precision $\lambda$ grows.

**Model Architecture**  To ensure that the improvements in sample quality on ImageNet arise from BSI as a method and not from the architecture of the underlying model, we have also trained U-Nets on ImageNet32. Table 3 shows that the U-Net exhibits the same characteristics as the DiT that we trained in Section 6.1, i.e. equivalent likelihoods between BFN, VDM and BSI with a consistent improvement in FID. We chose the U-Net parameterization of (Kingma et al., 2023), which is also listed in Appendix E.

*Table 3.* Trained with U-Net architecture.

| Model | BPD | FID |
|-------|-----|-----|
| BFN | $3.505 \pm 0.001$ | $14.2 \pm 0.4$ |
| VDM | $3.527 \pm 0.009$ | $15.4 \pm 1.5$ |
| BSI | $3.510 \pm 0.009$ | $12.8 \pm 0.6$ |

**Initial Precision**  In Fig. 7, we evaluate the impact of the initial precision $\lambda_0$ on likelihood and sample quality. While the likelihood of test data improves with falling $\lambda_0$, i.e. increasing initial noise, the sample quality depends on the number of sampling steps. For a large number of steps, larger $\lambda_0$ perform slightly better, but with fewer steps an intermediate $\lambda_0$ is preferred. With fewer total sampling steps, decreasing $\lambda_0$ ensures that the sampling process still spends enough steps in the intermediate noise range, which is responsible for the generation of large-scale features in the images (Rissanen et al., 2022).

**Training with $\mathcal{L}_M^\infty$**  By default, we train by optimizing the measurement loss $\mathcal{L}_M^\infty$ of the infinite-step ELBO, but in practice the model will only use finitely many steps. Fig. 8 shows that training on $\mathcal{L}_M^k$ does not confer a consistent advantage in sample quality or likelihood. This justifies training by optimizing $\mathcal{L}_M^\infty$ regardless of the number of steps used later and eliminates $k$ as a hyperparameter.

## 7  Conclusion

We have introduced our generative model BSI through a novel perspective on generative modeling that frames sample generation as iterative Bayesian inference. We have derived an ELBO for both finite steps and the infinite step limit and an importance sampling distribution to minimize the training loss variance. In addition, we have thoroughly discussed how BSI relates to BFN and DMs and shown that BSI includes BFN as a special case. Our experiments have demonstrated that BSI generates better samples than both VDM and BFN while achieving equivalent log-likelihoods on established image datasets. Overall, BSI contributes a Bayesian perspective to the landscape of probabilistic generative modeling that is theoretically simple and empirically effective.

**Software**

For our results, we rely on excellent software packages, notably `numpy` (Harris et al., 2020), `pytorch` (Paszke et al., 2019), `einops` (Rogozhnikov, 2022), `matplotlib` (Hunter, 2007), `h5py` (Collette, 2013), `hydra` (Yadan, 2019) and `jupyter` (Granger & Pérez, 2021).

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

# A  How BSI relates to . . .

## A.1  Bayesian Flow Networks

BFNs are a recent class of generative models for continuous and discrete data motivated from an information-theoretic perspective (Graves et al., 2023). In it, a sender communicates a latent sample to a receiver while trying to minimize the transported data volume. The sender compresses the data with entropy coding, so that minimizing the data volume is equivalent to the receiver maximizing the log-likelihood of the latent sample based on the information that it has received from the sender so far. Finally, a sample can be generated when the receiver also assumes the role of the sender and repeatedly refines its belief.

*Our generative approach in Section 3 includes BFN for continuous data as a special case.* To see this, we begin by choosing our belief prior $p(\boldsymbol{\mu}_0)$ as $\mathcal{N}_P(\mathbf{0}, \gamma_0)$ and letting $\gamma_0 \to \infty$, i.e. the initial belief mean will always be $\boldsymbol{\mu}_0 = \mathbf{0}$. With Lemma 3.4, this gives us

$$q(\boldsymbol{\mu}_\lambda \mid \boldsymbol{x}, \lambda) = \mathcal{N}_P\left(\frac{\lambda - \lambda_0}{\lambda}\boldsymbol{x}, \frac{\lambda^2}{\lambda - \lambda_0}\right). \tag{20}$$

If we now define $\alpha = \lambda - \lambda_0$, choose the initial precision $\lambda_0 = 1$ and write the Normal distribution in variance format, we see that

$$q(\boldsymbol{\mu}_\lambda \mid \boldsymbol{x}, \lambda) = \mathcal{N}\left(\frac{\alpha}{1 + \alpha}\boldsymbol{x}, \frac{\alpha}{(1 + \alpha)^2}\right), \tag{21}$$

which equals the BFN flow distribution $p_F(\boldsymbol{\theta} \mid \boldsymbol{x}; t)$ (Graves et al., 2023, Equation (76)) if we parametrize $\lambda$ (and therefore $\alpha$) in terms of $t \in [0, 1]$ as in Section 5.2.

Since a comprehensive description of BFN would go beyond the scope of this work, we will only point out the correspondence between terms from Section 3 and their BFN counterparts without explaining them in detail. For a complete description, we refer the reader to the original work (Graves et al., 2023).

The current belief $(\boldsymbol{\mu}_i, \lambda_i)$ is equivalent to the input distribution $p_I$ (Graves et al., 2023, Equation (43)). Lemma 2.1 is the equivalent of the Bayesian update function $h$ (Graves et al., 2023, Section 4.2). A noisy measurement $\boldsymbol{y} \sim \mathcal{N}_P(\boldsymbol{x}, \alpha)$ corresponds to the sender distribution $p_S$ (Graves et al., 2023, Equation (86)), while a noisy measurement $\boldsymbol{y} \sim \mathcal{N}_P(\hat{\boldsymbol{x}}, \alpha)$ of the model's current prediction $\hat{\boldsymbol{x}}$ of the true sample corresponds to the receiver distribution $p_R$ (Graves et al., 2023, Equation (88)). The output distribution $p_O$ and the Bayesian update distribution $p_U$ are just intermediate terms to derive the model and appear neither in the final training nor sampling algorithm.

Fixing the initial belief to $\boldsymbol{\mu}_0 = 0$ with infinite precision for BFN recovers the behavior described by Graves et al. (2023, Figures 3 and 4) and shown in Eq. (21) that the precision $(1 + \alpha)^2/\alpha$ of the flow / encoding distribution $q(\boldsymbol{\mu}_\lambda \mid \boldsymbol{x}, \lambda)$ in the ELBO first falls and then rises again as $\alpha$ grows. In contrast, with our belief prior $p(\boldsymbol{\mu}_0) = \mathcal{N}_P(\mathbf{0}, \lambda_0)$ of the same precision as the initial belief $(\boldsymbol{\mu}_0, \lambda_0)$ as we choose it in Section 3.2, the precision of $q(\boldsymbol{\mu}_\lambda \mid \boldsymbol{x}, \lambda)$ grows linearly in $\lambda$ (and $\alpha$) in lockstep with the precision of the belief $(\boldsymbol{\mu}_i, \lambda_i)$. We hypothesize that this makes learning for the model easier, because the noise level in its input varies linearly instead of non-linearly across noise levels. Furthermore, in BSI, the first sampling step will already contribute to drawing a random sample, since the initial input $\boldsymbol{\mu}_0$ to $f_\theta$ is random. In BFN, the initial belief is fixed to $\mathbf{0}$, which makes the first sampling step deterministic and equal across all samples.

In Section 3.2, we have argued that the reasonable range of prior precisions $\gamma_0$ is $[\lambda_0, \infty]$. *BSI and BFN occupy the two extremes of this range* with BSI using the least informed prior $\gamma_0 = \lambda_0$, i.e. making the fewest assumptions, and BFN the most informed one $\gamma_0 = \infty$. Note that these extremes are the only choices in the reasonable range for which the precision $\lambda^2(\lambda - \lambda_0 + \lambda_0^2/\gamma_0)^{-1}$ of the encoder q in Lemma 3.4 simplifies, i.e. to just $\lambda$ for BSI and $\lambda^2(\lambda - \lambda_0)^{-1}$ for BFN.

In our comparison to DMs in Appendix A.2, we see that BSI and BFN also differ in their associated noising process. While BSI's noising process, i.e. how one could go from a more precise measurement back to a less precise one, does not form a Markov chain, BFN's does, making BFN more similar to DMs.

In Appendix A.2, we exploit that BFN can be represented as a special case of BSI to derive a Markovian forward process for BFN as DMs.

## A.2 Diffusion Models

DMs are a widely used class of generative models built on the concept of inverting a diffusion process (Sohl-Dickstein et al., 2015; Ho et al., 2020). Given a sample $\boldsymbol{x}$, they define a Markov chain of increasingly noisy versions $\boldsymbol{x}_1, \boldsymbol{x}_2, \ldots$ of $\boldsymbol{x}$ where $\boldsymbol{x}_0 = \boldsymbol{x}$ and

$$\mathrm{p}(\boldsymbol{x}_i \mid \boldsymbol{x}_{i-1}) = \mathcal{N}(\alpha_i \boldsymbol{x}_{i-1}, \beta_i) \tag{22}$$

for some coefficients $\alpha_i$ and $\beta_i$. In training, a model learns to invert this Markov chain, which lets you finally generate data by sampling from a noise distribution and stepping along the learned, reverse Markov chain until you reach the data distribution.

While DMs initially achieved prominence in image generation (Dhariwal & Nichol, 2021), they have since been applied successfully across a variety of domains, such as text-to-image mapping (Saharia et al., 2022), fluid simulations (Lienen et al., 2024; Saydemir et al., 2024), adversarial attacks (Kollovieh et al., 2024a), temporal (Lüdke et al., 2023) and general point processes (Lüdke et al., 2024), molecular dynamics (Lewis et al., 2025), molecular structure generation (Ayadi et al., 2024), and time series forecasting (Kollovieh et al., 2023; 2024b).

DMs and BSI are remarkably similar at first glance. Both revolve around the concept of iteratively transforming noise into data samples, though DMs work with Langevin dynamics and BSI uses posterior inference. For training, both models aim to align a parametric distribution $\mathrm{p}_{\boldsymbol{\theta}}(\boldsymbol{x}'' \mid \boldsymbol{x}')$ with a distribution $\mathrm{q}(\boldsymbol{x}'' \mid \boldsymbol{x}', \boldsymbol{x})$ that describes a less noisy version $\boldsymbol{x}''$ of a noisy sample $\boldsymbol{x}'$ given that the true sample is $\boldsymbol{x}$.

However, conceptually, they approach sampling from two different perspectives. DMs start with the so-called forward process, where signal is iteratively converted into noise forming a Markov chain of intermediate states as in Eq. (22). Then, they revert this chain to derive the reverse process that enriches noise with data. In contrast, BSI defines the reverse process directly in the form of Lemma D.1 and never uses the associated forward process directly.

We can revert BSI's process to derive its "noising" process. This will let us see what BSI would look like as a DM and thus understand the relationship between the two. Assume that our current belief is $(\boldsymbol{\mu}, \lambda = \lambda_0 + \alpha)$ and we want to denoise further based on a sample $\boldsymbol{x}$ and measurement precision $\alpha'$, i.e. update our belief to $(\boldsymbol{\mu}', \lambda' = \lambda_0 + \alpha + \alpha')$. The denoising process described by Lemma D.1 tells us that

$$\mathrm{p}(\boldsymbol{\mu}' \mid \boldsymbol{\mu}, \boldsymbol{x}) = \mathcal{N}_{\mathrm{P}}\left(\tfrac{1}{\lambda'}\left[\lambda\boldsymbol{\mu} + \alpha'\boldsymbol{x}\right], \tfrac{\lambda'^2}{\alpha'}\right). \tag{23}$$

To find the noising process, we revert this and get

$$\mathrm{p}(\boldsymbol{\mu} \mid \boldsymbol{\mu}', \boldsymbol{x}) = \mathcal{N}\left(\xi^{-1}\left[\frac{\lambda\lambda'}{\alpha'}\boldsymbol{\mu}' + \lambda\left(\frac{\alpha}{\alpha + \lambda_0^2/\gamma_0} - 1\right)\boldsymbol{x}\right], \xi\right) \tag{24}$$

where $\xi = \lambda^2\left(\left(\alpha + \lambda_0^2/\gamma_0\right)^{-1} + \alpha'^{-1}\right)$ and $\gamma_0$ is the precision of the initial belief prior $\mathrm{p}(\boldsymbol{\mu}_0) = \mathcal{N}(\boldsymbol{0}, \gamma_0)$. Find the proof at the end of this section.

Plugging in $\gamma_0 = \lambda_0$, we get that the noising process of BSI is

$$\mathrm{p}(\boldsymbol{\mu} \mid \boldsymbol{\mu}', \boldsymbol{x}) = \mathcal{N}\left(\xi^{-1}\left[\frac{\lambda\lambda'}{\alpha'}\boldsymbol{\mu}' - \lambda_0\boldsymbol{x}\right], \xi\right) \quad \text{where} \quad \xi = \lambda\left(1 + \frac{\lambda}{\alpha'}\right). \tag{25}$$

Note that this distributions depends on $\boldsymbol{x}$ since $\lambda_0 > 0$. Therefore, BSI's forward process would not be Markov, i.e. you cannot add more noise to a belief state without knowing the sample $\boldsymbol{x}$ that the belief state originated from. While DMs with non-Markov forward processes exist (Song et al., 2021a; Chen et al., 2024), they are uncommon. In conclusion, we see that BSI can be represented as a DM, though with a rather complex, non-Markovian forward process.

As we have shown in Appendix A.1, BFN are a special case of our generative framework in Section 3 if we choose $\gamma_0 = \infty$. Curiously, Eq. (24) shows that this is the only prior on $\boldsymbol{\mu}_0$ for which the associated forward process is Markov as the coefficient of $\boldsymbol{x}$ becomes 0. This agrees with Xue et al. (2024), who have shown that BFN admit a formulation based on stochastic differential equations (SDEs), like score-based DMs.

*Proof of Eq.* (24). We know from Lemma 3.4 that

$$\mathrm{q}(\boldsymbol{\mu} \mid \boldsymbol{x}, \lambda) = \mathcal{N}_{\mathrm{P}}\left(\frac{\lambda - \lambda_0}{\lambda}\,\boldsymbol{x}, \frac{\lambda^2}{\lambda - \lambda_0 + \lambda_0^2/\gamma_0}\right) = \mathcal{N}_{\mathrm{P}}\left(\frac{\alpha}{\lambda}\,\boldsymbol{x}, \frac{\lambda^2}{\alpha + \lambda_0^2/\gamma_0}\right) \tag{26}$$

and from Lemma D.1 that

$$\mathrm{p}(\boldsymbol{\mu}' \mid \boldsymbol{\mu}, \boldsymbol{x}) = \mathcal{N}_{\mathrm{P}}\left(1/\lambda'\left[\lambda\boldsymbol{\mu} + \alpha'\boldsymbol{x}\right], \lambda'^2/\alpha'\right). \tag{27}$$

Therefore, $\mathrm{p}(\boldsymbol{\mu}, \boldsymbol{\mu}' \mid \boldsymbol{x})$ is a Gaussian linear system and we can use (Murphy, 2012, Equation (4.125)) to see that

$$\mathrm{p}(\boldsymbol{\mu} \mid \boldsymbol{\mu}', \boldsymbol{x}) = \mathcal{N}_{\mathrm{P}}(\boldsymbol{\nu}, \xi) \tag{28}$$

with

$$\xi = \lambda^2\left(\alpha + \frac{\lambda_0^2}{\gamma_0}\right)^{-1} + \left(\frac{\lambda}{\lambda'}\right)^2\frac{\lambda'^2}{\alpha'} = \lambda^2\left((\alpha + \lambda_0^2/\gamma_0)^{-1} + \alpha'^{-1}\right) \tag{29}$$

and

$$\boldsymbol{\nu} = \xi^{-1}\left[\frac{\lambda}{\lambda'}\frac{\lambda'^2}{\alpha'}\left(\boldsymbol{\mu}' - \frac{\alpha'}{\lambda'}\boldsymbol{x}\right) + \lambda^2(\alpha + \lambda_0^2/\gamma_0)^{-1}\,\alpha/\lambda\,\boldsymbol{x}\right] \tag{30}$$

$$= \xi^{-1}\left[\frac{\lambda\lambda'}{\alpha'}\boldsymbol{\mu}' + \lambda\left(\frac{\alpha}{\alpha + \lambda_0^2/\gamma_0} - 1\right)\boldsymbol{x}\right]. \tag{31}$$

$\square$

## A.3 Stochastic Interpolants

Stochastic interpolants are a broad class of continuous-time stochastic processes that can interpolate between any two probability distributions $\rho_0$ and $\rho_1$ (Albergo et al., 2025). They also prescribe how to learn the interpolants' dynamics to construct generative models and it is instructive to see how they relate to BSI. The subclass of *spatially linear one-sided interpolants* assumes that $\rho_0$ is a standard Normal distribution and defines the interpolant

$$\boldsymbol{x}_t = \alpha(t)\boldsymbol{z}_t + \beta(t)\boldsymbol{x}_1 \tag{32}$$

where $\boldsymbol{x}_1 \sim \rho_1$ and $\boldsymbol{z}_t \sim \mathcal{N}(\mathbf{0}, \boldsymbol{I})$. Given that $\alpha$ and $\beta$ are smooth, non-negative functions with $\alpha(0) = \beta(1) = 1$ and $\alpha(1) = \beta(0) = 0$, $\boldsymbol{x}_t$ smoothly interpolates between a standard Normal and the data distribution $\rho_1$.

For BSI, we can interpret the belief mean $\boldsymbol{\mu}_\lambda$ for a data sample $\boldsymbol{x}$

$$\boldsymbol{\mu}_\lambda = \frac{1}{\sqrt{\lambda}}\boldsymbol{z} + \frac{\lambda - \lambda_0}{\lambda}\boldsymbol{x} \tag{33}$$

as an interpolant that is normally distributed with mean $\mathbf{0}$ and precision $\lambda_0$ at $\lambda = \lambda_0$ and equals the data sample $\boldsymbol{x}$ at $\lambda = \infty$. We could rewrite this as an interpolant in the above sense on $[0, 1]$ by parameterizing $\lambda$ as a strictly increasing function $\lambda(t) : [0, 1] \to \mathbb{R}_+$ with $\lambda(0) = \lambda_0$ and $\lim_{t \to 1} \lambda(t) = \infty$, e.g. $\lambda(t) = \lambda_0 - \log(1 - t)$, similar to the mapping between score-based diffusion and stochastic interpolants (Albergo et al., 2025, Section 5.1). But to avoid the scaling and correction factors, we will consider it an interpolant on $[\lambda_0, \infty]$ instead with $\alpha(\lambda) = 1/\sqrt{\lambda}$ and $\beta(\lambda) = (\lambda - \lambda_0)/\lambda$. We will furthermore write $\alpha(\lambda)$ and $\beta(\lambda)$ as $\alpha$ and $\beta$ to reduce visual clutter.

(Albergo et al., 2025, Section 4.4) shows that the probability path $\rho(\lambda, \boldsymbol{\mu}_\lambda)$ of the interpolant solves the transport equation $\partial_\lambda \rho + \nabla \cdot (b\rho) = 0$ with the velocity field

$$b(\lambda, \boldsymbol{\mu}) = \frac{\dot{\alpha}}{\alpha}\boldsymbol{\mu} + \left(\dot{\beta} - \frac{\dot{\alpha}}{\alpha}\beta\right)\eta(\lambda, \boldsymbol{\mu}) \tag{34}$$

and its score $\nabla \log \rho(\lambda, \boldsymbol{\mu})$ is given by

$$s(\lambda, \boldsymbol{\mu}) = -\frac{\boldsymbol{\mu} - \beta\eta(\lambda, \boldsymbol{\mu})}{\alpha^2}. \tag{35}$$

$\eta(\lambda, \boldsymbol{\mu}) = \mathbb{E}[\boldsymbol{x} \mid \boldsymbol{\mu}_\lambda = \boldsymbol{\mu}]$ is the denoiser, i.e. the expected data sample that led to belief $\boldsymbol{\mu}$ at precision $\lambda$. Note that $\eta(\lambda, \boldsymbol{\mu})$ is learned with a model $\hat{\eta}$, which is equivalent to $f_{\boldsymbol{\theta}}(\boldsymbol{\mu}, \lambda)$ in BSI and fit by minimizing (Albergo et al., 2025, Eq. (4.21))

$$\mathcal{L}(\hat{\eta}) = \int_{\lambda_0}^{\lambda_M} \mathbb{E}\left[\frac{1}{2}\|\hat{\eta}(\lambda, \boldsymbol{\mu})\|_2^2 - \boldsymbol{\mu} \cdot \hat{\eta}(\lambda, \boldsymbol{\mu})\right] d\lambda. \tag{36}$$

$\mathcal{L}(\hat{\eta})$ is equivalent to $\mathcal{L}_M^\infty$ in Theorem 3.2 up to a constant factor and offset.

With the velocity field and score, we can write down the forward SDE corresponding to the probability path $\rho$ as (Albergo et al., 2025, Corollary 18)

$$d\boldsymbol{\mu}_\lambda = b_F(\lambda, \boldsymbol{\mu}_\lambda) d\lambda + \sqrt{2\varepsilon(\lambda)} dW_\lambda \tag{37}$$

where $W_\lambda$ is Brownian motion, $\varepsilon(\lambda) : \mathbb{R} \to \mathbb{R}_+$ is any noise level specification and

$$\begin{aligned} b_F(\lambda, \boldsymbol{\mu}) &= b(\lambda, \boldsymbol{\mu}) + \varepsilon(\lambda)s(\lambda, \boldsymbol{\mu}) \\ &= \left(\frac{\dot{\alpha}}{\alpha} - \frac{\varepsilon}{\alpha^2}\right)\boldsymbol{\mu} + \left(\frac{\varepsilon}{\alpha^2} - \frac{\dot{\alpha}}{\alpha} + \frac{\dot{\beta}}{\beta}\right)\beta\eta(\lambda, \boldsymbol{\mu}) \end{aligned} \tag{38}$$

is the forward drift. If we plug in $\alpha$ and $\beta$, we get

$$d\boldsymbol{\mu}_\lambda = \left[-\left(\frac{1}{2\lambda} + \varepsilon\lambda\right)\boldsymbol{\mu}_\lambda + \left(\varepsilon\lambda + \frac{1}{2\lambda} + \frac{\lambda_0}{\lambda(\lambda - \lambda_0)}\right)\frac{\lambda - \lambda_0}{\lambda}\eta(\lambda, \boldsymbol{\mu}_\lambda)\right] d\lambda + \sqrt{2\varepsilon} dW_\lambda. \tag{39}$$

Since this holds for any non-negative $\varepsilon$, we can choose $\varepsilon = \frac{1}{2\lambda^2}$ to simplify the equation to

$$d\boldsymbol{\mu}_\lambda = \frac{1}{\lambda}\left[\eta(\lambda, \boldsymbol{\mu}_\lambda) - \boldsymbol{\mu}_\lambda\right] d\lambda + \frac{1}{\lambda}dW_\lambda. \tag{40}$$

We can now sample from the learned stochastic interpolant by integrating Eq. (40) from $\lambda_0$ to $\lambda_M$ (Albergo et al., 2025, Algorithm 5). Let's say we are at precision $\lambda$ with state $\boldsymbol{\mu}_\lambda$ and want to move ahead by a step of length $\alpha$. With the Euler-Maruyama method suggested by (Albergo et al., 2025), the integration step becomes

$$\begin{aligned} \boldsymbol{\mu}_{\lambda+\alpha} &= \boldsymbol{\mu}_\lambda + \alpha \cdot \frac{1}{\lambda}\left[\hat{\eta}(\lambda, \boldsymbol{\mu}_\lambda) - \boldsymbol{\mu}_\lambda\right] + \frac{\sqrt{\alpha}}{\lambda}\boldsymbol{\varepsilon} \\ &= \frac{\lambda - \alpha}{\lambda}\boldsymbol{\mu}_\lambda + \frac{\alpha}{\lambda}\hat{\eta}(\lambda, \boldsymbol{\mu}_\lambda) + \frac{\sqrt{\alpha}}{\lambda}\boldsymbol{\varepsilon} \end{aligned} \tag{41}$$

where $\boldsymbol{\varepsilon} \sim \mathcal{N}(\boldsymbol{0}, \boldsymbol{I})$. This is almost the same as the BSI sampling step

$$\begin{aligned} \boldsymbol{\mu}_i &= \frac{\lambda_{i-1}\boldsymbol{\mu}_{i-1} + \alpha_i\left(\hat{\boldsymbol{x}}_{i-1} + \sqrt{\frac{1}{\alpha_i}}\boldsymbol{\varepsilon}_i\right)}{\lambda_{i-1} + \alpha_i} \\ &= \frac{\lambda_i - \alpha_i}{\lambda_i}\boldsymbol{\mu}_{i-1} + \frac{\alpha_i}{\lambda_i}\hat{\boldsymbol{x}}_{i-1} + \frac{\sqrt{\alpha_i}}{\lambda_i}\boldsymbol{\varepsilon}_i \end{aligned} \tag{42}$$

in Algorithm 3. The subtle difference is that Eq. (41) uses the current $\lambda$ to compute $\boldsymbol{\mu}_{\lambda+\alpha}$ whereas Eq. (42) uses the next $\lambda_i = \lambda_{i-1} + \alpha_i$ for $\boldsymbol{\mu}_i$. This difference reflects that BSI employs an exact Bayesian update rather than a first-order Euler-Maruyama approximation.

In summary, we can write BSI as a spatially linear one-sided stochastic interpolant, though two important differences remain. First, while the sampling steps Eqs. (41) and (42) are equivalent in the continuous limit of $\alpha \to 0$, they differ in practice due to their derivation from an SDE discretization and posterior inference, respectively. Second, stochastic interpolants require the interpolation's endpoints to equal the noise and data distribution exactly, which corresponds to $\lambda = \infty$ in the above formulation. In contrast, BSI only infers the sample $\boldsymbol{x}$ up to a maximum precision of $\lambda_M$, e.g. precisely enough to identify the exact color in an image with 8-bit color channels.

# B  ELBO in Bits per Dimension

A common metric in probabilistic modeling is the negative log-likelihood of unseen data. The benefits of this metric are that it is theoretically motivated by the probabilistic framework and it can be computed across domains regardless of data modality. If the negative log-likelihood is small, the generative model assigns high likelihood to the unseen data and can thus be regarded as a good model (though likelihood and sample quality are not necessarily the same thing (Theis et al., 2016)). For models that come with an ELBO like BSI, we can use it to upper bound the negative log-likelihood to compare against other ELBO-based or exact-likelihood models.

The negative log-likelihood is usually reported in bits per pixel, per color channel or, in general, per dimension. This unit comes from the fact that an entropy coder could use the model to encode samples $\boldsymbol{x} \in \mathbb{S}^d$ from a finite symbol alphabet $\mathbb{S}$ from the data distribution asymptotically in $-\log_2 p_{\boldsymbol{\theta}}(\boldsymbol{x})/d$ bits per dimension (Duda et al., 2015). Note that the underlying space $\mathbb{S}$ must be discrete. If it were continuous, $p_{\boldsymbol{\theta}}(\boldsymbol{x})$ would be a density and the theory would predict that we could compress $\boldsymbol{x}$ into a negative number of bits.

The discreteness requirement is a natural fit for many domains. While, for example, images are usually treated as tensors with continuous color values, the colors are actually stored as discrete values in the range $[0, 2^8 - 1]$ for 8-bit images. Similarly, audio data is a sequence of discrete values in, for example, a 16-bit range.

Let's say that $\mathbb{S}$ is the set of integers $\{0, \ldots, r-1\}$. Then we can compute an upper bound on the bits needed to encode $\boldsymbol{x} \in \mathbb{S}^d$ by

$$- \log_2 p(\boldsymbol{x}) \leq \log(2)(\mathcal{L}'_{\mathrm{R}} + \mathcal{L}^{\infty}_{\mathrm{M}}) \tag{43}$$

as per Theorems 3.1 and 3.2. The multiplication by $\log(2)$ converts the logarithms in $\mathcal{L}'_{\mathrm{R}}$ and $\mathcal{L}^{\infty}_{\mathrm{M}}$ to base 2. $\mathcal{L}'_{\mathrm{R}}$ is the same as $\mathcal{L}_{\mathrm{R}}$ but with a discretized Normal likelihood to account for the discrete nature of $\boldsymbol{x}$, i.e.

$$\mathcal{L}'_{\mathrm{R}} = \mathbb{E}_{\mathrm{q}(\boldsymbol{\mu}_{\lambda_{\mathrm{M}}} | \boldsymbol{x}, \lambda_{\mathrm{M}})} \left[ -\log \mathcal{N}'_{\mathrm{P}}(\boldsymbol{x} \mid \hat{\boldsymbol{x}}_{\lambda_{\mathrm{M}}}, \alpha_{\mathrm{R}}) \right] \tag{44}$$

where

$$\mathcal{N}'_{\mathrm{P}}(x_j \mid \hat{\boldsymbol{x}}_{\lambda_{\mathrm{M}}}, \alpha_{\mathrm{R}}) = \Phi(r_j \mid \hat{\boldsymbol{x}}_{\lambda_{\mathrm{M}}}, \alpha_{\mathrm{R}}) - \Phi(l_j \mid \hat{\boldsymbol{x}}_{\lambda_{\mathrm{M}}}, \alpha_{\mathrm{R}}). \tag{45}$$

$\Phi(r_j \mid \hat{\boldsymbol{x}}_{\lambda_{\mathrm{M}}}, \alpha_{\mathrm{R}})$ is the CDF of $\mathcal{N}(\hat{\boldsymbol{x}}_{\lambda_{\mathrm{M}}}, \alpha_{\mathrm{R}})$ and $l_j$ and $r_j$ are the boundaries of the discretization interval containing $x_j$, i.e.

$$l_j = \begin{cases} -\infty & \text{if } x_j < \frac{1}{2} \\ r - \frac{3}{2} & \text{if } x_j \geq r - \frac{3}{2} \\ \lfloor x_j - \frac{1}{2} \rfloor + \frac{1}{2} & \text{otherwise} \end{cases} \quad \text{and} \quad r_j = \begin{cases} \infty & \text{if } x_j \geq r - \frac{3}{2} \\ \frac{1}{2} & \text{if } x_j < \frac{1}{2} \\ \lfloor x_j + \frac{1}{2} \rfloor - \frac{1}{2} & \text{otherwise.} \end{cases} \tag{46}$$

$\mathcal{L}^{\infty}_{\mathrm{M}}$ is usually not discretized during ELBO computation as the latent variables only enter as a mean squared error instead of a log-likelihood. In a practical implementation, the latent variable distributions would need to be discretized as well, decreasing the ELBO slightly (Kingma et al., 2023; Townsend et al., 2019). If $\boldsymbol{x}$ is discretized to a different set of discrete symbols, e.g. numbers between $-1$ and $1$ instead of the integers $\mathbb{S}$, the boundaries of the discretization intervals and bin widths in the discretized Normal distribution have to be adapted accordingly.

# C  Preconditioning Derivation

We will assume in this section that the data is normalized such that $\mathbb{E}[\boldsymbol{x}] = \boldsymbol{0}$ and $\mathrm{Var}[\boldsymbol{x}] = \boldsymbol{I}$.

Assume that we have a current belief $(\boldsymbol{\mu}, \lambda)$. We derive the parameters $c_{\mathrm{skip}}$, $c_{\mathrm{out}}$ and $c_{\mathrm{in}}$ of the preconditioned model

$$f_{\boldsymbol{\theta}}(\boldsymbol{\mu}, \lambda) = c_{\mathrm{skip}} \boldsymbol{\mu} + c_{\mathrm{out}} f'_{\boldsymbol{\theta}}(c_{\mathrm{in}} \boldsymbol{\mu}, \lambda) \tag{47}$$

analogously to Karras et al. (2022). However, while we proceed in the same way, the resulting parameters for BSI differ from Karras et al. (2022) because BSI is not included in the family of DMs that Karras et al. (2022) consider, see Appendix A.2.

First, we require that $\mathrm{Var}_{\boldsymbol{x}}[c_{\mathrm{in}}\boldsymbol{\mu}] = \boldsymbol{I}$ for all $\lambda$. We know from Corollary 3.5 that

$$q(\boldsymbol{\mu} \mid \boldsymbol{x}, \lambda) = \mathcal{N}_{\mathrm{P}}\big((\lambda-\lambda_0)/\lambda\,\boldsymbol{x}, \lambda\big). \tag{48}$$

Therefore, $\mathrm{p}(\boldsymbol{x}, \boldsymbol{\mu})$ is a Gaussian linear system and (Murphy, 2012, Equation (4.126)) tells us that the variance of the marginal distribution of $\boldsymbol{\mu}$ is

$$\mathrm{Var}_{\boldsymbol{x}}[\boldsymbol{\mu}] = \left(\lambda^{\text{-}1} + \frac{(\lambda - \lambda_0)^2}{\lambda^2}\right)\boldsymbol{I}. \tag{49}$$

By plugging this into our requirement

$$\mathrm{Var}_{\boldsymbol{x}}[c_{\mathrm{in}}\boldsymbol{\mu}] = c_{\mathrm{in}}^2\,\mathrm{Var}_{\boldsymbol{x}}[\boldsymbol{\mu}] = \boldsymbol{I}, \tag{50}$$

we get immediately that

$$c_{\mathrm{in}} = \left(\lambda^{\text{-}1} + \frac{(\lambda - \lambda_0)^2}{\lambda^2}\right)^{\text{-}1/2} = \Big(\underbrace{1 + \frac{(\lambda - \lambda_0)^2}{\lambda}}_{=:\kappa}\Big)^{\text{-}1/2}\lambda^{1/2} = \sqrt{\lambda/\kappa}. \tag{51}$$

Next, we want to have the actual prediction target of $f'_{\boldsymbol{\theta}}$ during training to have unit variance, too. In training, we optimize the ELBO from Theorem 3.2, which comes down to minimizing

$$\big\|\boldsymbol{x} - f_{\boldsymbol{\theta}}(\boldsymbol{\mu}, \lambda)\big\|_2^2 \tag{52}$$

up to constant factors only depending on $\lambda$. If we plug in Eq. (47) and isolate $f'_{\boldsymbol{\theta}}$, this distance becomes

$$\big\|\boldsymbol{x} - c_{\mathrm{skip}}\boldsymbol{\mu} - c_{\mathrm{out}}f'_{\boldsymbol{\theta}}(c_{\mathrm{in}}\boldsymbol{\mu}, \lambda)\big\|_2^2 = c_{\mathrm{out}}^2\big\|f'_{\boldsymbol{\theta}}(c_{\mathrm{in}}\boldsymbol{\mu}, \lambda) - c_{\mathrm{out}}^{\text{-}1}(\boldsymbol{x} - c_{\mathrm{skip}}\boldsymbol{\mu})\big\|_2^2. \tag{53}$$

From this, we identify $c_{\mathrm{out}}^{\text{-}1}(\boldsymbol{x} - c_{\mathrm{skip}}\boldsymbol{\mu})$ as the actual training target for $f'_{\boldsymbol{\theta}}$. For the rest of this derivation, we denote use the shorthand $\alpha = \lambda - \lambda_0$ for the measurement precision accumulated in our belief $(\boldsymbol{\mu}, \lambda)$. After Corollary 3.5, we can write $\boldsymbol{\mu}$ as $\alpha/\lambda\,\boldsymbol{x} + \boldsymbol{z}$ where $\boldsymbol{z} \sim \mathcal{N}_{\mathrm{P}}(\boldsymbol{0}, \lambda)$ and find that the variance of the training target is

$$\begin{aligned}
\mathrm{Var}_{\boldsymbol{x}, \boldsymbol{z}}[c_{\mathrm{out}}^{\text{-}1}(\boldsymbol{x} - c_{\mathrm{skip}}\boldsymbol{\mu})] &= c_{\mathrm{out}}^{\text{-}2}\,\mathrm{Var}_{\boldsymbol{x}, \boldsymbol{z}}\left[\boldsymbol{x} - c_{\mathrm{skip}}\Big(\frac{\alpha}{\lambda}\,\boldsymbol{x} + \boldsymbol{z}\Big)\right] \\
&= c_{\mathrm{out}}^{\text{-}2}\,\mathrm{Var}_{\boldsymbol{x}, \boldsymbol{z}}\left[\Big(1 - c_{\mathrm{skip}}\frac{\alpha}{\lambda}\Big)\boldsymbol{x} - c_{\mathrm{skip}}\boldsymbol{z}\right] \\
&= c_{\mathrm{out}}^{\text{-}2}\left[\Big(1 - c_{\mathrm{skip}}\frac{\alpha}{\lambda}\Big)^2 + c_{\mathrm{skip}}^2\lambda^{\text{-}1}\right]\boldsymbol{I}
\end{aligned} \tag{54}$$

If we now require the effective training target to have unit variance, we see that

$$c_{\mathrm{out}}^2 = \Big(1 - c_{\mathrm{skip}}\frac{\alpha}{\lambda}\Big)^2 + c_{\mathrm{skip}}^2\lambda^{\text{-}1} = \left[1 + \frac{\alpha^2}{\lambda}\right]\frac{1}{\lambda}c_{\mathrm{skip}}^2 - 2\frac{\alpha}{\lambda}c_{\mathrm{skip}} + 1. \tag{55}$$

Following Karras et al. (2022), we now choose $c_{\mathrm{skip}}$ to minimize the impact of errors in the output of $f'_{\boldsymbol{\theta}}$ by minimizing the magnitude of $c_{\mathrm{out}}$. $c_{\mathrm{out}}^2$ is a polynomial in $c_{\mathrm{skip}}$ with positive leading coefficient, so we can find the minimizer as the root of

$$\frac{1}{2}\frac{\mathrm{d}c_{\mathrm{out}}^2}{\mathrm{d}c_{\mathrm{skip}}} = \left[1 + \frac{\alpha^2}{\lambda}\right]\frac{1}{\lambda}c_{\mathrm{skip}} - \frac{\alpha}{\lambda}, \tag{56}$$

which is at

$$c_{\mathrm{skip}} = \left[1 + \frac{\alpha^2}{\lambda}\right]^{\text{-}1}\alpha = \kappa^{\text{-}1}\alpha = \frac{\alpha}{\kappa}. \tag{57}$$

Finally, we can plug $c_{\mathrm{skip}}$ into Eq. (55) to get

$$c_{\mathrm{out}}^2 = \kappa\kappa^{\text{-}2}\frac{a^2}{\lambda} - 2\frac{\alpha}{\lambda}\kappa^{\text{-}1}\alpha + 1 = \kappa^{\text{-}1}\left(\frac{a^2}{\lambda} - 2\frac{\alpha^2}{\lambda} + \left[1 + \frac{\alpha^2}{\lambda}\right]\right) = \kappa^{\text{-}1} \tag{58}$$

and consequently $c_{\mathrm{out}} = \kappa^{\text{-}1/2} = \sqrt{1/\kappa}$.

## D  Proofs

### D.1  Proof of Theorem 3.1

We will begin with some auxiliary insights. First, we consider the marginal distribution of the updated belief $(\boldsymbol{\mu}', \lambda')$. This means that our current belief about a sample $\boldsymbol{x}$ is $(\boldsymbol{\mu}, \lambda)$ and now we want to know the distribution of $\boldsymbol{\mu}'$ after updating $\boldsymbol{\mu}$ with Lemma 2.1 marginalized over all possible noisy measurements $\boldsymbol{y}$ with precision $\alpha$. Note that $\lambda'$ is deterministic as it neither depends on $\boldsymbol{x}$ nor $\boldsymbol{y}$.

**Lemma D.1** (Update Marginal). *Let $\boldsymbol{x}, \boldsymbol{\mu} \in \mathbb{R}^n$ and $\lambda, \alpha \in \mathbb{R}_+$. Then the distribution of the posterior belief mean $\boldsymbol{\mu}'$ marginalized over all measurements $\boldsymbol{y}$ made with precision $\alpha$ is*

$$\mathrm{p}(\boldsymbol{\mu}' \mid \boldsymbol{\mu}, \boldsymbol{x}, \alpha) = \mathop{\mathbb{E}}_{\boldsymbol{y} \sim \mathcal{N}_{\mathrm{P}}(\boldsymbol{x}, \alpha \boldsymbol{I})} \big[ \mathrm{p}(\boldsymbol{\mu}' \mid \boldsymbol{\mu}, \boldsymbol{x}, \alpha, \boldsymbol{y}) \big] = \mathcal{N}_{\mathrm{P}}\big( {}^1\!/\!\lambda' \big[ \lambda \boldsymbol{\mu} + \alpha \boldsymbol{x} \big], {\lambda'}^2\!/\!\alpha \big). \tag{59}$$

*Proof.* The noisy measurement is a Normal random variable $\boldsymbol{y} \sim \mathcal{N}_{\mathrm{P}}(\boldsymbol{x}, \alpha)$ and the mean of our posterior belief $(\boldsymbol{\mu}', \lambda')$ after observing $\boldsymbol{y}$ is the deterministic linear transformation

$$\boldsymbol{\mu}' = {}^1\!/\!\lambda' \left[ \lambda \boldsymbol{\mu} + \alpha \boldsymbol{y} \right] \tag{60}$$

of this random variable. The statement follows immediately by the linear transformation property of the Normal distribution. $\qquad \square$

From this, we can see that the update marginal from multiple intermediate measurements is the same as from a single measurement with the combined precision of the intermediate measurements.

**Lemma D.2.** *Let $\boldsymbol{x}, \boldsymbol{\mu}, \boldsymbol{\mu}', \boldsymbol{\mu}'' \in \mathbb{R}^n$ and $\lambda, \alpha, \alpha' \in \mathbb{R}_+$. $\boldsymbol{\mu}'$ is the posterior belief mean after a measurement with precision $\alpha$ and $\boldsymbol{\mu}''$ the posterior belief mean after a second, subsequent measurement with precision $\alpha'$. Then we have that the marginal distribution of the second update is*

$$\mathop{\mathbb{E}}_{\mathrm{p}(\boldsymbol{\mu}' \mid \boldsymbol{\mu}, \boldsymbol{x}, \alpha)} [\mathrm{p}(\boldsymbol{\mu}'' \mid \boldsymbol{\mu}', \boldsymbol{x}, \alpha')] = \mathrm{p}(\boldsymbol{\mu}'' \mid \boldsymbol{\mu}, \boldsymbol{x}, \alpha + \alpha'). \tag{61}$$

*Proof.* We know from Lemma D.1 that $\boldsymbol{\mu}'$ is a random variable

$$\mathrm{p}(\boldsymbol{\mu}' \mid \boldsymbol{\mu}, \boldsymbol{x}, \alpha) = \mathcal{N}_{\mathrm{P}}\big( \underbrace{{}^1\!/\!\lambda' \big[ \lambda \boldsymbol{\mu} + \alpha \boldsymbol{x} \big]}_{=: \boldsymbol{\nu}}, \underbrace{{\lambda'}^2\!/\!\alpha}_{=: \xi} \big) \tag{62}$$

and $\boldsymbol{\mu}''$ is a random variable that depends linearly on $\boldsymbol{\mu}'$

$$\mathrm{p}(\boldsymbol{\mu}'' \mid \boldsymbol{\mu}', \boldsymbol{x}, \alpha') = \mathcal{N}_{\mathrm{P}}\big( {}^1\!/\!\lambda'' \big[ \lambda' \boldsymbol{\mu}' + \alpha' \boldsymbol{x} \big], {\lambda''}^2\!/\!\alpha' \big). \tag{63}$$

As such, they jointly form a Gaussian linear system for which the marginal distribution of $\boldsymbol{\mu}''$ is (Murphy, 2012, Equation (4.126))

$$\mathop{\mathbb{E}}_{\mathrm{p}(\boldsymbol{\mu}' \mid \boldsymbol{\mu}, \boldsymbol{x}, \alpha)} [\mathrm{p}(\boldsymbol{\mu}'' \mid \boldsymbol{\mu}', \boldsymbol{x}, \alpha')] = \mathcal{N}\left( {}^1\!/\!\lambda'' \big[ \lambda' \boldsymbol{\nu} + \alpha' \boldsymbol{x} \big], \frac{\alpha'}{{\lambda''}^2} + \frac{{\lambda'}^2}{{\lambda''}^2 \xi} \right). \tag{64}$$

Plugging $\boldsymbol{\nu}$ into the mean expression and simplifying yields the marginal mean

$$ {}^1\!/\!\lambda'' \big[ \lambda \boldsymbol{\mu} + (\alpha + \alpha') \boldsymbol{x} \big]. \tag{65}$$

Similarly, plugging $\xi$ into the covariance expression and simplifying yields the marginal covariance

$$\frac{\alpha + \alpha'}{{\lambda''}^2}. \tag{66}$$

If we now recall from Lemma 2.1 that

$$\lambda' = \lambda + \alpha \quad \text{and} \quad \lambda'' = \lambda' + \alpha' = \lambda + \alpha + \alpha', \tag{67}$$

we can identify Eq. (64) as $\mathrm{p}(\boldsymbol{\mu}'' \mid \boldsymbol{\mu}, \boldsymbol{x}, \alpha + \alpha')$. $\qquad \square$

This trivially generalizes to any finite sequence of measurements, which can be collapsed into a single measurement with the total precision instead.

We will furthermore need to know the KL divergence between the update marginal distributions of the same belief but based on two different samples $\boldsymbol{x}$ and $\boldsymbol{x}'$.

**Lemma D.3.** *Let $\boldsymbol{x}, \boldsymbol{x}', \boldsymbol{\mu} \in \mathbb{R}^n$ and $\lambda, \alpha \in \mathbb{R}_+$. Then*

$$D_{\mathrm{KL}}(\mathrm{p}(\boldsymbol{\mu}' \mid \boldsymbol{\mu}, \boldsymbol{x}, \alpha), \mathrm{p}(\boldsymbol{\mu}' \mid \boldsymbol{\mu}, \boldsymbol{x}', \alpha)) = {}^1\!/{}_2\, \alpha \|\boldsymbol{x} - \boldsymbol{x}'\|_2^2. \tag{68}$$

*Proof.* Both update marginal distributions – with $\boldsymbol{x}$ and $\boldsymbol{x}'$ – are Normal distributions of equal precision $\xi := \frac{\lambda'^2}{\alpha}$ as given by Lemma D.1 and respective means of

$$\boldsymbol{\nu} = {}^1\!/{}_{\lambda'}\big[\lambda\boldsymbol{\mu} + \alpha\boldsymbol{x}\big] \quad \text{and} \quad \boldsymbol{\nu}' = {}^1\!/{}_{\lambda'}\big[\lambda\boldsymbol{\mu} + \alpha\boldsymbol{x}'\big]. \tag{69}$$

As a consequence, the closed form solution for the KL divergence between two equal-covariance Normal distributions becomes

$$
\begin{aligned}
D_{\mathrm{KL}}(\mathrm{p}(\boldsymbol{\mu}' \mid \boldsymbol{\mu}, \boldsymbol{x}, \alpha), \mathrm{p}(\boldsymbol{\mu}' \mid \boldsymbol{\mu}, \boldsymbol{x}', \alpha)) &= \frac{1}{2}(\boldsymbol{\nu} - \boldsymbol{\nu}')^{\mathsf{T}} \xi (\boldsymbol{\nu} - \boldsymbol{\nu}') \\
&= \frac{1}{2}(\boldsymbol{x} - \boldsymbol{x}')^{\mathsf{T}} \alpha {\lambda'}^{-1} \xi {\lambda'}^{-1} \alpha (\boldsymbol{x} - \boldsymbol{x}') \\
&= \frac{1}{2}(\boldsymbol{x} - \boldsymbol{x}')^{\mathsf{T}} \alpha (\boldsymbol{x} - \boldsymbol{x}') \\
&= \frac{1}{2}\alpha \|\boldsymbol{x} - \boldsymbol{x}'\|_2^2
\end{aligned}
\tag{70}
$$

$\square$

Equipped with these, we can derive the ELBO.

**Theorem 3.1.** *Let $\boldsymbol{x} \in \mathbb{R}^n$ and $\alpha_{\mathrm{R}}, \alpha_i \in \mathbb{R}_+, i \in [k]$. Then the log-likelihood of $\boldsymbol{x}$ is lower-bounded as*

$$\log \mathrm{p}(\boldsymbol{x}) \geq -\mathcal{L}_{\mathrm{R}} - \mathcal{L}_{\mathrm{M}}^k \tag{3}$$

*by a reconstruction term $\mathcal{L}_{\mathrm{R}}$ and a measurement term $\mathcal{L}_{\mathrm{M}}^k$,*

$$\mathcal{L}_{\mathrm{R}} = \mathop{\mathbb{E}}_{\mathrm{q}(\boldsymbol{\mu}_k \mid \boldsymbol{x}, \lambda_k)} \big[ {-}\log \mathcal{N}_{\mathrm{P}}(\boldsymbol{x} \mid \hat{\boldsymbol{x}}_k, \alpha_{\mathrm{R}}) \big] \quad \text{and} \quad \mathcal{L}_{\mathrm{M}}^k = \frac{k}{2} \mathop{\mathbb{E}}_{\substack{i \sim \mathcal{U}(0, k-1) \\ \mathrm{q}(\boldsymbol{\mu}_i \mid \boldsymbol{x}, \lambda_i)}} \big[ \alpha_{i+1} \|\boldsymbol{x} - \hat{\boldsymbol{x}}_i\|_2^2 \big] \tag{4}$$

*where*

$$\mathrm{q}(\boldsymbol{\mu}_i \mid \boldsymbol{x}, \lambda_i) = \mathop{\mathbb{E}}_{\mathrm{p}(\boldsymbol{\mu}_0)} \big[ \mathrm{p}(\boldsymbol{\mu}_i \mid \boldsymbol{\mu}_0, \boldsymbol{x}, \lambda_i) \big], \quad \hat{\boldsymbol{x}}_i = f_{\boldsymbol{\theta}}(\boldsymbol{\mu}_i, \lambda_i) \quad \text{and} \quad \lambda_i = \lambda_0 + \sum_{j=1}^{i} \alpha_j. \tag{5}$$

*Proof.* For any distribution $\mathrm{p}(\boldsymbol{x})$ and any latent variable $\boldsymbol{z}$, i.e. any choice of prior $\mathrm{p}(\boldsymbol{z})$, encoding distribution $\mathrm{q}(\boldsymbol{z} \mid \boldsymbol{x})$ and likelihood $\mathrm{p}(\boldsymbol{x} \mid \boldsymbol{z})$, we have the variational lower bound

$$\log \mathrm{p}(\boldsymbol{x}) \geq - \mathop{\mathbb{E}}_{\mathrm{q}(\boldsymbol{z} \mid \boldsymbol{x})} [- \log \mathrm{p}(\boldsymbol{x} \mid \boldsymbol{z})] - D_{\mathrm{KL}}(\mathrm{q}(\boldsymbol{z} \mid \boldsymbol{x}), \mathrm{p}(\boldsymbol{z})) \tag{71}$$

on $\log \mathrm{p}(\boldsymbol{x})$ (Kingma & Welling, 2013). In particular, we can choose our sequence of beliefs as the latent variable $\boldsymbol{z} = \{\boldsymbol{\mu}_0, \dots, \boldsymbol{\mu}_k\}$ and define the likelihood of $\boldsymbol{x}$ under this latent variable as

$$\mathrm{p}(\boldsymbol{x} \mid \boldsymbol{z}) = \mathcal{N}_{\mathrm{P}}(\boldsymbol{x} \mid \hat{\boldsymbol{x}}_k, \alpha_{\mathrm{R}}). \tag{72}$$

Remember that $\hat{\boldsymbol{x}}_k = f_{\boldsymbol{\theta}}(\boldsymbol{\mu}_k, \lambda_k)$ is the model's estimate of $\boldsymbol{x}$.

Since the belief means $\boldsymbol{\mu}_1, \ldots, \boldsymbol{\mu}_k$ are updated only based on their predecessor after Lemma 2.1, they form a Markov chain conditional on $\boldsymbol{x}$ and we can write the encoding distribution as

$$q(\boldsymbol{z} \mid \boldsymbol{x}) = p(\boldsymbol{\mu}_0) \prod_{i=1}^{k} p(\boldsymbol{\mu}_i \mid \boldsymbol{\mu}_{i-1}, \boldsymbol{x}, \alpha_i). \tag{73}$$

Each $p(\boldsymbol{\mu}_i \mid \boldsymbol{\mu}_{i-1}, \boldsymbol{x}, \alpha_i)$ is the update marginal of $\boldsymbol{\mu}_{i-1}$ over all possible noisy measurements of $\boldsymbol{x}$ with precision $\alpha_i$ from Lemma D.1. Our encoding distribution is ignorant about the influence of $\boldsymbol{x}$ on the initial belief $\boldsymbol{\mu}_0$, because there is no closed form for $p(\boldsymbol{\mu}_0 \mid \boldsymbol{x})$. Since we can choose any encoding, not encoding $\boldsymbol{x}$ in $\boldsymbol{\mu}_0$ at all is valid.

If we now plug Eq. (73) into the first term of Eq. (71), we get

$$\mathbb{E}_{q(\boldsymbol{z}|\boldsymbol{x})}[-\log p(\boldsymbol{x} \mid \boldsymbol{z})] = \mathbb{E}_{p(\boldsymbol{\mu}_0)} \mathbb{E}_{p(\boldsymbol{\mu}_1|\boldsymbol{\mu}_0,\boldsymbol{x},\alpha_1)} \cdots \mathbb{E}_{p(\boldsymbol{\mu}_k|\boldsymbol{\mu}_{k-1},\boldsymbol{x},\alpha_k)}[-\log p(\boldsymbol{x} \mid \boldsymbol{z})]. \tag{74}$$

The intermediate expectations collapse into a single measurement with the sum of all precisions $\bar{\alpha}_i = \sum_{j=1}^{i} \alpha_j$ according to Lemma D.2, because $\boldsymbol{\mu}_1, \ldots, \boldsymbol{\mu}_{k-1}$ do not appear in the inner log-likelihood, and we are left with

$$\mathbb{E}_{q(\boldsymbol{z}|\boldsymbol{x})}[-\log p(\boldsymbol{x} \mid \boldsymbol{z})] = \mathbb{E}_{p(\boldsymbol{\mu}_0)} \mathbb{E}_{p(\boldsymbol{\mu}_k|\boldsymbol{\mu}_0,\boldsymbol{x},\bar{\alpha}_k)}[-\log p(\boldsymbol{x} \mid \boldsymbol{z})]. \tag{75}$$

Since $\lambda_i = \lambda_0 + \sum_{j=1}^{i} \alpha_j = \lambda_0 + \bar{\alpha}_i$, we can define

$$p(\boldsymbol{\mu}_k \mid \boldsymbol{\mu}_0, \boldsymbol{x}, \lambda) \coloneqq p(\boldsymbol{\mu}_k \mid \boldsymbol{\mu}_0, \boldsymbol{x}, \alpha = \lambda - \lambda_0) = p(\boldsymbol{\mu}_k \mid \boldsymbol{\mu}_0, \boldsymbol{x}, \bar{\alpha}_k). \tag{76}$$

If we now define

$$q(\boldsymbol{\mu}_k \mid \boldsymbol{x}, \lambda_k) \coloneqq \mathbb{E}_{p(\boldsymbol{\mu}_0)}\left[p(\boldsymbol{\mu}_k \mid \boldsymbol{\mu}_0, \boldsymbol{x}, \lambda_k)\right], \tag{77}$$

we can rewrite Eq. (75) as

$$\mathbb{E}_{q(\boldsymbol{z}|\boldsymbol{x})}[-\log p(\boldsymbol{x} \mid \boldsymbol{z})] = \mathbb{E}_{q(\boldsymbol{\mu}_k|\boldsymbol{x},\lambda_k)}[-\log p(\boldsymbol{x} \mid \boldsymbol{\mu}_k)] \tag{78}$$

which equals the definition of $\mathcal{L}_{\mathrm{R}}$ after plugging in Eq. (72).

Next, we investigate the KL-divergence in Eq. (71). We begin by defining the latent prior $p(\boldsymbol{z})$ autoregressively as

$$p(\boldsymbol{z}) = p(\boldsymbol{\mu}_0) \prod_{i=1}^{k} p(\boldsymbol{\mu}_i \mid \boldsymbol{\mu}_{i-1}, \hat{\boldsymbol{x}}_{i-1}, \alpha_i) \tag{79}$$

where $\hat{\boldsymbol{x}}_{i-1} = f_{\boldsymbol{\theta}}(\boldsymbol{\mu}_{i-1}, \lambda_{i-1})$ is the point estimate of $\boldsymbol{x}$ produced by our model based on the belief at step $i-1$. So the prior for $\boldsymbol{\mu}_i$ is the update marginal in Lemma D.1 if $\hat{\boldsymbol{x}}_{i-1}$ were the actual sample $\boldsymbol{x}$.

Now, we plug Eqs. (73) and (79) into the KL-divergence term from Eq. (71).

$$\begin{aligned}
D_{\mathrm{KL}}(q(\boldsymbol{z} \mid \boldsymbol{x}), p(\boldsymbol{z})) &= \mathbb{E}_{q(\boldsymbol{z}|\boldsymbol{x})}\left[\log \frac{q(\boldsymbol{z} \mid \boldsymbol{x})}{p(\boldsymbol{z})}\right] \\
&= \mathbb{E}_{q(\boldsymbol{z}|\boldsymbol{x})}\left[\log \frac{p(\boldsymbol{\mu}_0)}{p(\boldsymbol{\mu}_0)} + \sum_{i=1}^{k} \log \frac{p(\boldsymbol{\mu}_i \mid \boldsymbol{\mu}_{i-1}, \boldsymbol{x}, \alpha_i)}{p(\boldsymbol{\mu}_i \mid \boldsymbol{\mu}_{i-1}, \hat{\boldsymbol{x}}_{i-1}, \alpha_i)}\right] \\
&= \sum_{i=1}^{k} \mathbb{E}_{q(\boldsymbol{z}|\boldsymbol{x})}\left[\log \frac{p(\boldsymbol{\mu}_i \mid \boldsymbol{\mu}_{i-1}, \boldsymbol{x}, \alpha_i)}{p(\boldsymbol{\mu}_i \mid \boldsymbol{\mu}_{i-1}, \hat{\boldsymbol{x}}_{i-1}, \alpha_i)}\right] \\
&= \sum_{i=1}^{k} \mathbb{E}_{p(\boldsymbol{\mu}_0)} \mathbb{E}_{p(\boldsymbol{\mu}_1|\boldsymbol{\mu}_0,\boldsymbol{x},\alpha_1)} \cdots \mathbb{E}_{p(\boldsymbol{\mu}_i|\boldsymbol{\mu}_{i-1},\boldsymbol{x},\alpha_i)}\left[\log \frac{p(\boldsymbol{\mu}_i \mid \boldsymbol{\mu}_{i-1}, \boldsymbol{x}, \alpha_i)}{p(\boldsymbol{\mu}_i \mid \boldsymbol{\mu}_{i-1}, \hat{\boldsymbol{x}}_{i-1}, \alpha_i)}\right] \\
&= \sum_{i=1}^{k} \mathbb{E}_{q(\boldsymbol{\mu}_{i-1}|\boldsymbol{x},\lambda_{i-1})}\left[D_{\mathrm{KL}}\big(p(\boldsymbol{\mu}_i \mid \boldsymbol{\mu}_{i-1}, \boldsymbol{x}, \alpha_i), p(\boldsymbol{\mu}_i \mid \boldsymbol{\mu}_{i-1}, \hat{\boldsymbol{x}}_{i-1}, \alpha_i)\big)\right]
\end{aligned} \tag{80}$$

The intermediate expectations have collapsed again according to Lemma D.2 in the same way as for the reconstruction term.

We know the closed form for the inner KL divergences from Lemma D.3, so we can further simplify the KL-divergence term to

$$D_{\mathrm{KL}}(\mathrm{q}(\boldsymbol{z} \mid \boldsymbol{x}), \mathrm{p}(\boldsymbol{z})) = \frac{1}{2} \sum_{i=1}^{k} \mathop{\mathbb{E}}_{\mathrm{q}(\boldsymbol{\mu}_{i-1} \mid \boldsymbol{x}, \lambda_{i-1})} \left[ \alpha_i \|\boldsymbol{x} - \hat{\boldsymbol{x}}_{i-1}\|_2^2 \right]. \tag{81}$$

Shifting the sum indices by 1 and replacing the sum $\sum_{i=0}^{k-1}$ with $k \mathbb{E}_{i \sim \mathcal{U}(0, k-1)}$ yields $\mathcal{L}_{\mathrm{M}}^k$. $\qquad \square$

## D.2   Proof of Theorem 3.2

**Theorem 3.2.** *Let $\alpha_{\mathrm{R}}, \alpha_{\mathrm{M}} \in \mathbb{R}_+$. For any sequence of precision schedules $\alpha_{k,i}$ for $k \in \mathbb{N}, i \in [k]$ such that $\sum_{i=1}^{k} \alpha_{k,i} = \alpha_{\mathrm{M}}$ and the sequence of functions $[k] \to \mathbb{R}_+ : i \mapsto \alpha_{k,i}$ converges uniformly to 0, we can take the limit of Theorem 3.1 as $k \to \infty$ to get*

$$\mathcal{L}_{\mathrm{R}} = \mathop{\mathbb{E}}_{\mathrm{q}(\boldsymbol{\mu}_{\lambda_{\mathrm{M}}} \mid \boldsymbol{x}, \lambda_{\mathrm{M}})} \left[ -\log \mathcal{N}_{\mathrm{P}}(\boldsymbol{x} \mid \hat{\boldsymbol{x}}_{\lambda_{\mathrm{M}}}, \alpha_{\mathrm{R}}) \right] \quad and \quad \mathcal{L}_{\mathrm{M}}^\infty = \frac{\alpha_{\mathrm{M}}}{2} \mathop{\mathbb{E}}_{\substack{\lambda \sim \mathcal{U}(\lambda_0, \lambda_{\mathrm{M}}) \\ \mathrm{q}(\boldsymbol{\mu}_\lambda \mid \boldsymbol{x}, \lambda)}} \left[ \|\boldsymbol{x} - \hat{\boldsymbol{x}}_\lambda\|_2^2 \right] \tag{6}$$

*where $\mathrm{q}(\boldsymbol{\mu}_\lambda \mid \boldsymbol{x}, \lambda) = \mathbb{E}_{\mathrm{p}(\boldsymbol{\mu}_0)} \left[ \mathrm{p}(\boldsymbol{\mu}_\lambda \mid \boldsymbol{\mu}_0, \boldsymbol{x}, \lambda) \right]$, $\lambda_{\mathrm{M}} = \lambda_0 + \alpha_{\mathrm{M}}$ and $\hat{\boldsymbol{x}}_\lambda = f_{\boldsymbol{\theta}}(\boldsymbol{\mu}_\lambda, \lambda)$.*

*Proof.* Since $\mathcal{L}_{\mathrm{R}}$ only depends on $\sum_i \alpha_{k,i}$ but not individual $\alpha_{k,i}$, the equivalence of the finite and infinite step $\mathcal{L}_{\mathrm{R}}$ is immediately apparent.

For $\mathcal{L}_{\mathrm{M}}^k$, we will consider its sum form from Eq. (81).

$$\mathcal{L}_{\mathrm{M}}^k = \frac{1}{2} \sum_{i=1}^{k} \mathop{\mathbb{E}}_{\mathrm{q}(\boldsymbol{\mu}_{i-1} \mid \boldsymbol{x}, \lambda_{i-1})} \left[ \alpha_i \|\boldsymbol{x} - \hat{\boldsymbol{x}}_{i-1}\|_2^2 \right] = \frac{1}{2} \sum_{i=1}^{k} \alpha_i \underbrace{\mathop{\mathbb{E}}_{\mathrm{q}(\boldsymbol{\mu}_{i-1} \mid \boldsymbol{x}, \lambda_{i-1})} \left[ \|\boldsymbol{x} - \hat{\boldsymbol{x}}_{i-1}\|_2^2 \right]}_{=:h(\lambda_{i-1})} \tag{82}$$

Note that $h(\lambda_{i-1})$ is a deterministic function of $\lambda_{i-1}$ and $\lambda_0, \ldots, \lambda_k$ is a partition of the interval $[\lambda_0, \lambda_0 + \alpha_m] = [\lambda_0, \lambda_{\mathrm{M}}]$ with interval lengths of $\alpha_i$. It follows that Eq. (82) is a Riemann sum. Since $f_{\boldsymbol{\theta}}$ is a neural network, we can assume that $h(\lambda_{i-1})$ is continuous almost everywhere. Combined with the fact that the interval lengths $\{\alpha_i\}$ converge uniformly to 0, it follows that $\mathcal{L}_{\mathrm{M}}^k$ converges to the Riemann integral

$$\lim_{k \to \infty} \mathcal{L}_{\mathrm{M}}^k = \frac{1}{2} \int_{\lambda_0}^{\lambda_{\mathrm{M}}} \mathop{\mathbb{E}}_{\mathrm{q}(\boldsymbol{\mu}_\lambda \mid \boldsymbol{x}, \lambda)} \left[ \|(\boldsymbol{x} - \hat{\boldsymbol{x}}_\lambda)\|_2^2 \right] \mathrm{d}\lambda \tag{83}$$

as $k \to \infty$. It follows trivially that

$$\lim_{k \to \infty} \mathcal{L}_{\mathrm{M}}^k = \frac{\alpha_{\mathrm{M}}}{2} \int_{\lambda_0}^{\lambda_{\mathrm{M}}} \frac{1}{\alpha_{\mathrm{M}}} \mathop{\mathbb{E}}_{\mathrm{q}(\boldsymbol{\mu}_\lambda \mid \boldsymbol{x}, \lambda)} \left[ \|(\boldsymbol{x} - \hat{\boldsymbol{x}}_\lambda)\|_2^2 \right] \mathrm{d}\lambda \tag{84}$$

$$= \frac{\alpha_{\mathrm{M}}}{2} \mathop{\mathbb{E}}_{\substack{\lambda \sim \mathcal{U}(\lambda_0, \lambda_{\mathrm{M}}) \\ \mathrm{q}(\boldsymbol{\mu}_\lambda \mid \boldsymbol{x}, \lambda)}} \left[ \|(\boldsymbol{x} - \hat{\boldsymbol{x}}_\lambda)\|_2^2 \right] = \mathcal{L}_{\mathrm{M}}^\infty. \tag{85}$$

$\square$

## D.3   Proof of Lemma 3.3

**Lemma 3.3.** *If $h$ is strictly decreasing, $\mathcal{L}_{\mathrm{M}}^\infty < \mathcal{L}_{\mathrm{M}}^k$ for any $k$ and any precision schedule $\{\alpha_i\}$.*

*Proof.* In the proof of Theorem 3.2, we have established that $\mathcal{L}_{\mathrm{M}}^k$ is a Riemannian sum of $h$, where we evaluate $h$ on the most-negative edge of each interval. Since $h$ is a non-negative, strictly decreasing function, the

discretization error on the interval $[\lambda_{i-1}, \lambda_i]$

$$\epsilon := \alpha_i h(\lambda_{i-1}) - \int_{\lambda_{i-1}}^{\lambda_i} h(\lambda) \, d\lambda \tag{86}$$

is also non-negative. Now consider a refinement of the discretization with $\lambda' \in (\lambda_{i-1}, \lambda_i)$ and the post-refinement discretization error on that interval

$$\epsilon' := (\lambda' - \lambda_{i-1})h(\lambda_{i-1}) + (\lambda_i - \lambda')h(\lambda') - \int_{\lambda_{i-1}}^{\lambda_i} h(\lambda) \, d\lambda = (\lambda' - \lambda_{i-1} - \alpha_i)h(\lambda_{i-1}) + (\lambda_i - \lambda')h(\lambda') + \epsilon. \tag{87}$$

Next, we express $\epsilon'$ in terms of $\epsilon$ as

$$\begin{aligned} \epsilon' &= (\lambda' - \lambda_{i-1} - \alpha_i)h(\lambda_{i-1}) + (\lambda_i - \lambda')h(\lambda') + \epsilon \\ &= (\lambda' - \lambda_i)h(\lambda_{i-1}) + (\lambda_i - \lambda')h(\lambda') + \epsilon \\ &= (\lambda_i - \lambda')(h(\lambda') - h(\lambda_{i-1})) + \epsilon. \end{aligned} \tag{88}$$

We know that $(\lambda_i - \lambda') > 0$, because $\lambda' \in (\lambda_{i-1}, \lambda_i)$, and $(h(\lambda') - h(\lambda_{i-1})) < 0$, because $h$ is strictly decreasing. It follows that $\epsilon' < \epsilon$.

This means that any refinement of the ELBO with more steps reduces the non-negative error between the Riemannian sum $\mathcal{L}_M^k$ and its limit $\mathcal{L}_M^\infty$. In other words, $\mathcal{L}_M^\infty < \mathcal{L}_M^k$ for all $k$. $\qquad\square$

### D.4 Proof of Lemma 3.4 and Corollary 3.5

The ELBO in Theorems 3.1 and 3.2 has one part that looks like it might not be so straightforward: the encoding distribution $q(\boldsymbol{\mu}_\lambda \mid \boldsymbol{x}, \lambda)$. Its definition contains a marginalization over the belief prior $p(\boldsymbol{\mu}_0)$, which we still need to specify. Let's see what $q(\boldsymbol{\mu}_\lambda \mid \boldsymbol{x}, \lambda)$ becomes if we choose a zero-mean, isotropic Normal prior $p(\boldsymbol{\mu}_0)$.

**Lemma 3.4.** *Let* $\lambda_0, \gamma_0 \in \mathbb{R}_+$, $p(\boldsymbol{\mu}_0) = \mathcal{N}_P(\boldsymbol{0}, \gamma_0)$ *and* $\lambda \geq \lambda_0$. *Then*

$$q(\boldsymbol{\mu}_\lambda \mid \boldsymbol{x}, \lambda) = \mathcal{N}_P\left(\frac{\lambda - \lambda_0}{\lambda} \boldsymbol{x}, \frac{\lambda^2}{\lambda - \lambda_0 + \lambda_0^2/\gamma_0}\right). \tag{9}$$

*Proof.* Let $p(\boldsymbol{\mu}_\lambda \mid \boldsymbol{\mu}_0, \boldsymbol{x}, \lambda)$ be the marginal distribution of $\boldsymbol{\mu}_\lambda$ after a measurement of precision $\alpha = \lambda - \lambda_0$, i.e.

$$p(\boldsymbol{\mu}_\lambda \mid \boldsymbol{\mu}_0, \boldsymbol{x}, \lambda) = p(\boldsymbol{\mu}_\lambda \mid \boldsymbol{\mu}_0, \boldsymbol{x}, \alpha = \lambda - \lambda_0). \tag{89}$$

We know from Lemma D.1 that

$$p(\boldsymbol{\mu}_\lambda \mid \boldsymbol{\mu}_0, \boldsymbol{x}, \alpha = \lambda - \lambda_0) = \mathcal{N}_P\left(1/\lambda\left[\lambda_0 \boldsymbol{\mu}_0 + (\lambda - \lambda_0)\boldsymbol{x}\right], \lambda^2/(\lambda - \lambda_0)\right). \tag{90}$$

Since $p(\boldsymbol{\mu}_0)$ is also Gaussian and $\boldsymbol{\mu}_\lambda$ depends linearly on $\boldsymbol{\mu}_0$, they form a Gaussian linear system for which the marginal distribution of $\boldsymbol{\mu}_\lambda$ is (Murphy, 2012, Equation (4.126))

$$q(\boldsymbol{\mu}_\lambda \mid \boldsymbol{x}, \lambda) = \mathbb{E}_{p(\boldsymbol{\mu}_0)}\left[p(\boldsymbol{\mu}_\lambda \mid \boldsymbol{\mu}_0, \boldsymbol{x}, \lambda)\right] = \mathcal{N}\left(1/\lambda\left[\lambda_0 \boldsymbol{0} + (\lambda - \lambda_0)\boldsymbol{x}\right], \frac{\lambda - \lambda_0}{\lambda^2} + \frac{\lambda_0^2}{\lambda^2 \gamma_0}\right). \tag{91}$$

By pulling $\lambda^{-2}$ out of the covariance and inverting to get a precision, we get the claimed result. $\qquad\square$

If we now choose $\gamma_0 = \lambda_0$, we get the simple BSI prior and the result ELBO encoder.

**Corollary 3.5.** *Let* $\lambda_0 \in \mathbb{R}_+$, $p(\boldsymbol{\mu}_0) \sim \mathcal{N}_P(\boldsymbol{0}, \lambda_0)$ *and* $\lambda \geq \lambda_0$. *Then*

$$q(\boldsymbol{\mu}_\lambda \mid \boldsymbol{x}, \lambda) = \mathcal{N}_P\left(\frac{\lambda - \lambda_0}{\lambda} \boldsymbol{x}, \lambda\right). \tag{10}$$

*Proof.* If we choose $\gamma_0 = \lambda_0$ in Lemma 3.4, we get

$$q(\boldsymbol{\mu}_\lambda \mid \boldsymbol{x}, \lambda) = \mathcal{N}_{\mathrm{P}}\left(\frac{\lambda - \lambda_0}{\lambda}\,\boldsymbol{x}, \frac{\lambda^2}{\lambda - \lambda_0 + \lambda_0^2/\lambda_0}\right). \tag{92}$$

The precision simplifies to

$$\frac{\lambda^2}{\lambda - \lambda_0 + \lambda_0^2/\lambda_0} = \frac{\lambda^2}{\lambda - \lambda_0 + \lambda_0} = \lambda, \tag{93}$$

proving the result. $\qquad\square$

### D.5  Proof of Corollary 3.6

**Corollary 3.6.** *Let* $\mathrm{p}(\lambda)$ *be a probability distribution with support* $[\lambda_0, \lambda_{\mathrm{M}}]$. *Then we have*

$$\mathcal{L}_{\mathrm{M}}^\infty = \frac{1}{2}\,\underset{\substack{\lambda \sim \mathrm{p}(\lambda) \\ q(\boldsymbol{\mu}_\lambda \mid \boldsymbol{x}, \lambda)}}{\mathbb{E}}\left[\frac{1}{\mathrm{p}(\lambda)}\|\boldsymbol{x} - \hat{\boldsymbol{x}}_\lambda\|_2^2\right]. \tag{11}$$

*Proof.* We know from Eq. (83) that $\mathcal{L}_{\mathrm{M}}^\infty$ is the following Riemann integral.

$$\mathcal{L}_{\mathrm{M}}^\infty = \frac{1}{2}\int_{\lambda_0}^{\lambda_{\mathrm{M}}}\underset{q(\boldsymbol{\mu}_\lambda \mid \boldsymbol{x}, \lambda)}{\mathbb{E}}\left[\|(\boldsymbol{x} - \hat{\boldsymbol{x}}_\lambda)\|_2^2\right]\mathrm{d}\lambda \tag{94}$$

Now we can trivially multiply by $\mathrm{p}(\lambda)/\mathrm{p}(\lambda)$ inside the expectation, proving the statement.

$$\mathcal{L}_{\mathrm{M}}^\infty = \frac{1}{2}\int_{\lambda_0}^{\lambda_{\mathrm{M}}}\underset{q(\boldsymbol{\mu}_\lambda \mid \boldsymbol{x}, \lambda)}{\mathbb{E}}\left[\frac{p(\lambda)}{p(\lambda)}\|(\boldsymbol{x} - \hat{\boldsymbol{x}}_\lambda)\|_2^2\right]\mathrm{d}\lambda \tag{95}$$

$$= \frac{1}{2}\int_{\lambda_0}^{\lambda_{\mathrm{M}}}\mathrm{p}(\lambda)\underset{q(\boldsymbol{\mu}_\lambda \mid \boldsymbol{x}, \lambda)}{\mathbb{E}}\left[\frac{1}{p(\lambda)}\|(\boldsymbol{x} - \hat{\boldsymbol{x}}_\lambda)\|_2^2\right]\mathrm{d}\lambda \tag{96}$$

$$\square$$

### D.6  Proof of Eq. (13)

*Proof.* We know from Corollary 3.5 that we can write $\boldsymbol{\mu}_\lambda = \lambda-\lambda_0/\lambda\,\boldsymbol{x} + 1/\sqrt{\lambda}\,\boldsymbol{\varepsilon}$ for Gaussian noise $\boldsymbol{\varepsilon} \sim \mathcal{N}(\boldsymbol{0}, \boldsymbol{I})$ independent of $\boldsymbol{x}$. Together with the assumption $f_{\boldsymbol{\theta}}(\boldsymbol{\mu}, \lambda) = \boldsymbol{\mu}$, we can rewrite $h$ as

$$\begin{aligned}
h(\lambda) &= \underset{q(\boldsymbol{\mu}_\lambda \mid \boldsymbol{x}, \lambda)}{\mathbb{E}}\|\boldsymbol{x} - \hat{\boldsymbol{x}}_\lambda\|_2^2 \\
&= \underset{\boldsymbol{\varepsilon} \sim \mathcal{N}(\boldsymbol{0}, \boldsymbol{I})}{\mathbb{E}}\left\|\boldsymbol{x} - \frac{\lambda - \lambda_0}{\lambda}\,\boldsymbol{x} + \frac{1}{\sqrt{\lambda}}\boldsymbol{\varepsilon}\right\|_2^2 \\
&= \underset{\boldsymbol{\varepsilon} \sim \mathcal{N}(\boldsymbol{0}, \boldsymbol{I})}{\mathbb{E}}\left\|\frac{\lambda_0}{\lambda}\boldsymbol{x} + \frac{1}{\sqrt{\lambda}}\boldsymbol{\varepsilon}\right\|_2^2 \\
&= \underset{\boldsymbol{\varepsilon} \sim \mathcal{N}(\boldsymbol{0}, \boldsymbol{I})}{\mathbb{E}}\left(\frac{\lambda_0}{\lambda}\right)^2\|\boldsymbol{x}\|_2^2 + \frac{1}{\lambda}\|\boldsymbol{\varepsilon}\|_2^2 - 2\frac{\lambda_0}{\sqrt{\lambda^3}}\boldsymbol{x}\cdot\boldsymbol{\varepsilon}
\end{aligned} \tag{97}$$

If we now make use of our assumption that $\mathbb{E}[\boldsymbol{x}] = \boldsymbol{0}$ and $\mathrm{Var}[\boldsymbol{x}] = \boldsymbol{I}$, we can distribute the expectation across terms and get

$$\underset{\boldsymbol{x}}{\mathbb{E}}[h(\lambda)] = \left(\frac{\lambda_0}{\lambda}\right)^2\underbrace{\underset{\boldsymbol{x}}{\mathbb{E}}\left[\|\boldsymbol{x}\|_2^2\right]}_{=n} + \frac{1}{\lambda}\underbrace{\underset{\boldsymbol{\varepsilon}}{\mathbb{E}}\left[\|\boldsymbol{\varepsilon}\|_2^2\right]}_{=n} - 2\frac{\lambda_0}{\sqrt{\lambda^3}}\underbrace{\underset{\boldsymbol{x},\boldsymbol{\varepsilon}}{\mathbb{E}}[\boldsymbol{x}\cdot\boldsymbol{\varepsilon}]}_{=0} \propto \frac{\lambda_0^2}{\lambda^2} + \frac{1}{\lambda}. \tag{98}$$

$$\square$$

# E   Experiment Details

*Table 4.* Test set log-likelihood on CIFAR10 of the same U-Net in different models.

| Model | Training Steps | BPD |
|-------|----------------|-----|
| VDM | 10 M | 2.65 |
| BSI | | 2.64 |
| BFN | 5 M | 2.66 |
| BSI | | 2.65 |

We trained each model on 4 H100 GPUs at a batch size of 128 on CIFAR10 and 512 on ImageNet32 and ImageNet64. Training progressed at about 26,300 steps per hour for the U-Net on CIFAR10 and 6,100 steps per hour for the DiT-L-2 backbones on ImageNet32. If we take the different batch sizes into account, the two model architectures needed about equal amounts of training time. Total training time for the 10 M step training on CIFAR10 came to about two weeks.

Furthermore, we take an exponential moving average (EMA) of model weights (Song et al., 2021b; Nichol & Dhariwal, 2021). We provide an overview of the model and training hyperparameters in Table 5, and show the U-Net and DiT parameters in Tables 6 to 9. On ImageNet32, we train the models with a cosine learning rate scheduler (with linear warm up from $1 \times 10^{-8}$) to achieve faster convergence. Note that we reduced the training steps to 100 k for our parameter studies to make them computationally feasible.

To reduce the variance of the training loss further, we use low-discrepancy sampling for $t$ in Algorithm 2 as proposed by Kingma et al. (2023). Instead of sampling $b$ independent $t$ for a batch size of $b$, we set $t_i = {}^{i-1}/b + \delta \mod 1, i \in [b]$ for a shared $\delta \sim \mathcal{U}(0,1)$ where $\mod 1$ means that we discard the integer part of the result. The marginal distribution of each $t_i$ is $\mathcal{U}(0,1)$, but jointly they cover the $[0,1]$ interval more uniformly than independent samples would, smoothing out the loss across batches.

*Table 5.* Model and training parameters of BSI on CIFAR10 and all three models on ImageNet32.

|  | Parameter | CIFAR10 | ImageNet32 (64) |
|---|-----------|---------|-----------------|
| BSI | $\alpha_0$ | | $1 \times 10^{-2}$ |
| | $\alpha_M$ | | $1 \times 10^6$ |
| | $\alpha_R$ | | $2 \times 10^6$ |
| Optim. | Learning rate | $2 \times 10^{-4}$ | $5 \times 10^{-4}$ |
| | LR Scheduler | None | Cosine $\downarrow 5 \times 10^{-5}$ |
| | Weight decay | | $1 \times 10^{-2}$ |
| | Batch size | 128 | 512 |
| | Steps | 10 M | 2 M (1 M) |
| EMA | $\beta$ | | 0.9999 |
| | First update after step | | 1000 |

*Table 6.* U-Net hyperparameters for CIFAR10.

| Parameter | Value |
|---|---|
| Hidden dim. | 128 |
| Levels | 32 |
| Dropout | 0.1 |
| Attention heads | 1 |
| Convolution padding | Zeros |

*Table 7.* DiT hyperparameters for ImageNet32.

| Parameter | Value |
|---|---|
| Architecture | DiT-L-2 |
| Hidden dim. | 1024 |
| Depth | 24 |
| Attention heads | 16 |
| Dropout | 0.05 |
| Patch Size | 2 |

*Table 8.* U-Net hyperparameters for ImageNet32.

| Parameter | Value |
|---|---|
| Hidden dim. | 256 |
| Levels | 32 |
| Dropout | 0.1 |
| Attention heads | 1 |
| Convolution padding | Zeros |

*Table 9.* DiT hyperparameters for ImageNet64.

| Parameter | Value |
|---|---|
| Architecture | DiT-L-4 |
| Hidden dim. | 1024 |
| Depth | 24 |
| Attention heads | 16 |
| Dropout | 0.05 |
| Patch Size | 4 |

## F Generated Samples

Fig. 9 shows generated samples from models trained on ImageNet32 for visual reference.

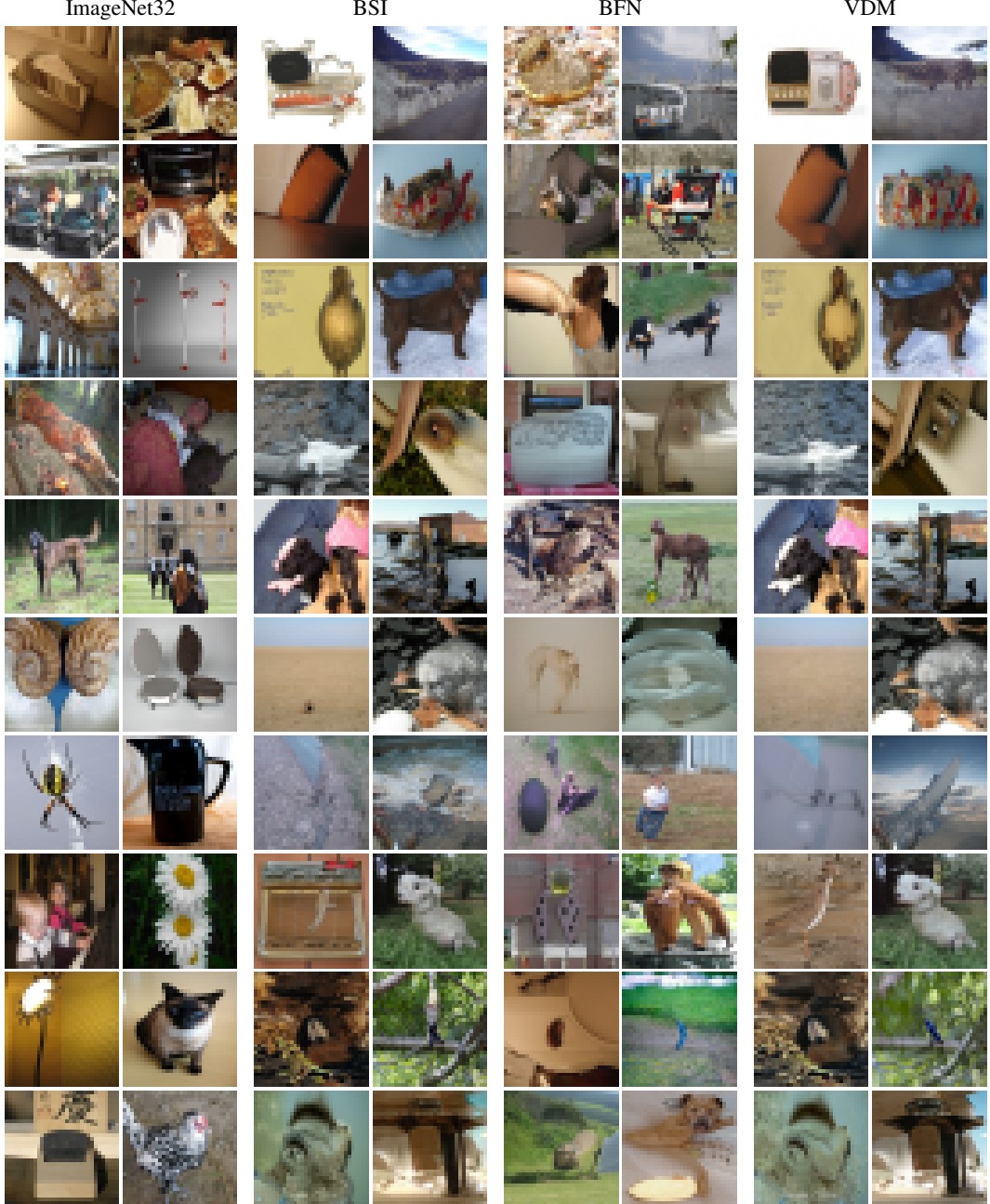

*Figure 9.* Samples from BSI, BFN and VDM trained on ImageNet32. Generated with 1024 steps. The first two columns show samples from the dataset for comparison.

