# OpenReview forum: "Generative Modeling with Bayesian Sample Inference"
_TMLR — Decision pending for TMLR_

### Review · Reviewer_zBq7 · 2026-05-27

**Summary Of Contributions:**

This paper proposes Bayesian Sample Inference (BSI). BSI is a generative model that treats data generation as a step-by-step Bayesian inference process using noisy predictions. The method uses a Gaussian distribution to track the belief about an unknown sample. It takes turns predicting the sample and updating the belief using noisy data. The authors also create finite-step and continuous (infinite-step) ELBO loss functions for training. Additionally, the paper introduces an importance-sampling method, discusses connections to diffusion models, and shows that Bayesian Flow Networks (BFNs) are just a special case of this framework.

In the experiments, the authors test BSI on ImageNet32, ImageNet64, and CIFAR10. They mainly compare it with BFNs and Variational Diffusion Models (VDMs). The main result is that BSI gets similar log-likelihood scores but achieves better sample quality (measured by FID) in these tests.

The paper is clearly written, and the Bayesian perspective is good. However, my main concern is that its overall framework is not very different from VDM. The paper itself notes that BSI, BFN, and VDM share a very similar ELBO structure. The only real differences are the choice of encoder, the initial belief distribution, and the sampling updates. Because of this, the technical and experimental contributions seem small. I am not convinced that BSI is a completely new generative model; rather, it looks like a minor variation of existing variational diffusion models.

**Audience:**

Yes

**Audience Explanation:**

Some members of the TMLR audience working on likelihood-based generative modeling, diffusion models, Bayesian Flow Networks, or theoretical unifications of generative models would likely find the Bayesian posterior-inference formulation interesting. The paper gives a clean perspective on BFN and VDM-like objectives, and the connection between Bayesian posterior updates and generative sampling may be useful pedagogically.

However, I expect the audience interest to be somewhat narrow unless the authors either reduce the novelty claims or provide stronger evidence that BSI leads to practically meaningful advantages. As currently presented, the work seems most valuable as a reinterpretation and modest extension of BFN/VDM rather than as a broadly impactful new generative modeling framework. The reported likelihood results are equivalent rather than improved, and the sample-quality improvements, while nontrivial, are not accompanied by enough ablations to show that the gains are intrinsic to the proposed framework.

Thus, I believe the paper may interest a subset of the TMLR audience, but the current manuscript needs stronger evidence or more carefully scoped claims to meet the standard for acceptance.

**Broader Impact Concerns:**

I do not see specific broader-impact concerns beyond those generally associated with unconditional image generative models, such as the possibility of generating synthetic images that could be misused. The paper appears to be methodological and does not introduce a new application domain with obvious additional ethical risks. A short broader impact statement acknowledging standard generative-model misuse risks would be sufficient.

**Claims And Evidence:**

No

**Claims Explanation:**

The paper provides clear math and a coherent probabilistic interpretation, with useful connections to BFN and VDM. However, several strong claims lack sufficient evidence:

* **Limited Novelty:** The overall framework is too similar to VDM. Because BSI, BFN, and VDM share a common ELBO structure, the differences (encoder, prior, and update rule) are minor. The paper does not prove this is a genuinely new modeling principle rather than a small modification to existing models.
* **Weak Likelihood Results:** For a paper focused on ELBO and likelihood optimization, BSI does not clearly improve NLL over baselines. The main improvements are only in FID, which weakens the core technical claim.
* **Unclear Source of FID Gains:** The better FID scores might simply come from specific hyperparameter choices, like the stochastic initial belief or initial precision. The paper lacks controlled baselines to show that a similarly tuned VDM or BFN would not achieve the same results.

Overall, BSI appears to be a well-written variant of VDM rather than a substantially novel generative model. Furthermore, the FID comparison is unfair. VDM is specifically optimized for NLL, and the VDM authors already noted that tuning the objective weights can significantly improve FID. By highlighting BSI's better FID against VDM's NLL-focused results, the paper is essentially comparing its own strength against the baseline's weakness.

**Requested Changes:**

- **Clarify and narrow the novelty claims**

The paper should more precisely state what is mathematically new relative to VDM, BFN, non-Markovian diffusion models, and stochastic-interpolant-style objectives. If the main novelty is a Bayesian reinterpretation plus a particular prior/encoder choice, the claims should be adjusted accordingly.


- **Strengthen the evidence for likelihood/NLL optimization**

Since the paper derives and emphasizes an ELBO, the authors should either demonstrate a clear likelihood advantage or explicitly reduce the claim that BSI improves likelihood-based modeling. Equivalent BPD is not, by itself, strong evidence for a new NLL optimization method.

- **Disentangle the effect of the initial prior precision**

The current evidence suggests that the stochastic initial belief and the choice of $\lambda_0$ may be central to the empirical gains. The authors should include ablations showing how much of the improvement remains when this factor is isolated from the rest of the BSI framework.
Add stronger ablations on inherited diffusion/VDM design components.
The paper should quantify the dependence on Fourier features, preconditioning, precision encoding, and architecture. Without such ablations, it is difficult to attribute improvements to the proposed framework rather than to existing diffusion-model engineering.

- **Ensure fair empirical comparisons**

Please compare BSI's FID only against baselines that are explicitly optimized for FID, and compare its NLL only against models optimized for NLL. Specifically, claiming superior FID by comparing against VDM (which is primarily optimized for NLL) is unfair. The comparisons need to be aligned with the baselines' actual optimization goals.

---

> ### Author Response · Authors · 2026-06-11
> **Rebuttal**
>
> Thank you for your review.
>
> **Likelihood claims**: We kindly want to point out that *we do not claim that BSI improves likelihoods*. We propose a new generative model from a probabilistic inference perspective that is trained via ELBO optimization. The equivalent likelihoods show that BSI is a competitive likelihood model while achieving better sample quality than similar likelihood-based models.
>
> **Novelty**: We respectfully disagree that BSI is a minor variation of VDMs. Table 1 and Section 4 clearly show the differences: BSI differs in the ELBO encoder, the sampling update and allows for a more general latent prior. Furthermore, we want to point out that the BSI update is trivial to implement numerically-stably, in contrast to VDM's. Regarding BFNs, we show that BSI generalizes them which is a non-trivial insight since BFNs are introduced and derived from a completely different, information-theoretic perspective. Additionally, our experiments show that BSI has a significant advantage in terms sample quality over BFNs specifically. The shared ELBO form is a unifying observation across the family, not evidence that the models are equivalent: VDMs, BFNs and BSI instantiate that form with materially different encoders, priors and update rules.
>
> **FID gains and tuning**: We want to highlight that *BSI is not tuned for FID* either. In fact, all three models are trained in the exact same way to make the comparison fair and the experiments as meaningful as possible, as we have explained in Section 6. The objective of our experiments was to examine these likelihood-based generative models using the exact same backbone architecture to understand the effects their theory has on likelihoods and sample quality. Therefore, our result that BSI improves sample quality against similar likelihood-based models across datasets and backbone architectures is reliable and meaningful.
>
> **Effect of initial prior precision**: We would like to point the reviewer to Figure 7, where we test exactly this relationship. It clearly shows that smaller $\lambda_0$ improve likelihood, while there is a U-shaped relationship with FID.
>
> **Effect of inherited components**: We trained the baselines ourselves with the exact same backbone architecture and training parameters as BSI. Therefore, any improvement we see over the baselines is necessarily due to the BSI construction, since that is the only difference.
>
> **Broader impact**: We will add a broader-impact statement on the standard misuse risks of generative models.

---

### Review · Reviewer_AekJ · 2026-05-27

**Summary Of Contributions:**

The paper proposes Bayesian Sample Inference (BSI), a generative model derived from iterative Gaussian posterior inference. The core idea is to treat sample generation as an inference problem: starting from a broad Gaussian belief, the model alternates between (i) predicting $\hat{x}$ from the current belief, (ii) drawing a noisy measurement, and (iii) performing a closed-form Bayesian update of the belief.

The concrete contributions are:

1. A new probabilistic formulation of iterative generative modeling, including a finite-step ELBO (Theorem 3.1) and its infinite-step counterpart (Theorem 3.2), together with a proof (Lemma 3.3) that the infinite-step ELBO is a tighter bound.
2. A log-uniform importance sampling scheme (Corollary 3.6 + Eq. 13) that reduces the Monte Carlo variance of the ELBO.
3. A unification result: BFN (Graves et al., 2023) is recovered as the special case of BSI. BSI's contribution over BFN is essentially the choice of an uninformed prior that allows non-trivial randomness.
4. A derivation of BSI as a diffusion model with a non-Markov forward process, and an articulation of how BSI relates to spatially linear one-sided stochastic interpolants (Appendix A.3).

Key strengths

1. The framework is clean and intuitive; the figures (especially Figs. 1 and 2) make the conceptual contribution accessible.
2. The theoretical work is rigorous and the proofs are well-presented.
3. The unification with BFN and the comparison table (Table 1) clarify the design space of this family of models, and the diffusion / stochastic-interpolant connections are useful for situating the work.

Key weaknesses

1. The log-likelihood improvements are within reported standard deviation, i.e., BSI is statistically tied with BFN/VDM on this metric.
2. The algorithmic delta from BFN is the choice of belief prior (plus the resulting changes that propagate from it). The paper is upfront about this, but it does mean the contribution is more conceptual/unifying than algorithmic.
3. BSI can be written as a specific spatially linear one-sided stochastic interpolant; the paper would benefit from a clearer statement of what BSI offers *beyond* that framework.
4. Experimental scope is restricted to small-resolution images and a narrow set of baselines (BFN, VDM); modern strong baselines (EDM, flow matching, consistency models) are absent.

**Audience:**

Yes

**Audience Explanation:**

Generative modeling is a central topic in the TMLR audience. The paper offers three things that are likely of interest to non-trivial subcommunities: (1) a Bayesian sample-inference perspective that simplifies and unifies the BFN derivation; (2) an explicit characterization of BFN as the special case; (3) a careful empirical comparison against VDM and BFN with matched architectures and training budgets.

**Broader Impact Concerns:**

I do not have specific broader impact concerns.

**Claims And Evidence:**

Yes

**Claims Explanation:**

The theoretical claims are supported by detailed and, as far as I checked, correct proofs in the appendices. The empirical claims are made in appropriately calibrated language. There are some places where claims could be sharpened, but I would not call any claim unsupported.

**Requested Changes:**

1. Compare against at least one modern strong baseline. VDM and BFN are appropriate primary baselines, but the absence of EDM, flow matching, or any score-based DM trained with current best practices makes it hard to assess whether the FID numbers are competitive in absolute terms. Even a single additional baseline at the same architecture and compute would substantially strengthen the paper.

2. The paper evaluates on 32×32 and 64×64 images only. Given the strong claims about a general-purpose Bayesian generative framework, a paragraph in the conclusion explicitly noting the limitation to small-resolution images, and whether scaling to e.g. ImageNet 256 or to non-image modalities is expected to preserve the observed FID gap, would be appropriate.

3. Report wall-clock or FLOPs cost of training and sampling for BSI vs. VDM vs. BFN to confirm the comparison is iso-compute.

---

> ### Author Response · Authors · 2026-06-11
> **Rebuttal**
>
> Thank you for your review and for the accurate summary of our contributions.
>
> **Log-likelihood**: BSI being tied with VDMs and BFNs in terms of log-likelihood mirrors exactly our claims. Importantly, produces better samples than these baselines despite the structural similarity, which we examine in our experiments.
>
> **Nature of the contribution**: It is correct that, in hindsight, BSI extends BFNs by a single hyperprior. We would stress two points. First, arriving at this is far from obvious: BFNs are derived from an information-theoretic sender/receiver perspective, whereas BSI arises from a Bayesian belief-inference perspective, and their equivalence is a result of our analysis rather than a starting assumption. Second, our experiments show that the hyperprior has a significant and consistent effect on sample quality across datasets and architectures.
>
> **Stochastic interpolants**: Appendix A.3 establishes this relationship and names two concrete differences. First, BSI performs an exact Bayesian posterior update rather than a first-order Euler-Maruyama discretization of the interpolant SDE. Second, stochastic interpolants require the endpoints to match the noise and data distributions exactly, i.e. infinite precision, whereas BSI infers the sample only up to a finite maximum precision, e.g. just enough to resolve 8-bit color.
>
> **More baselines**: Our experimental design deliberately isolates the effect of the BSI formulation. We compare against BFNs and VDMs because of their structural similarities (see Table 1 and Equation (14)), so any difference can be attributed to the underlying theory rather than to the architecture, sampler, or training recipe. EDM, flow matching and consistency models differ along several of these axes at once, which would confound exactly the comparison we want to make. As such, our experiments are plainly aimed at exploring the effects of the BSI derivation compared to similar models and not at establishing state of the art performance.
>
> **Higher resolution images**: Like any diffusion-like model known to us, higher resolutions would require working in a latent space. We deliberately chose to stay in data space to avoid the added complexity of an auto-encoder and to minimize the effect of further components on our experimental evaluation. We will extend the conclusion to state that higher-resolution image generation would require latent-space modeling.
>
> **Compute equivalence**: The comparison is iso-compute by construction modulo the difference in formulas in Table 1. All three models use identical architectures (DiT-L-2 and DiT-L-4, and the VDM U-Net), training budgets (1M, 2M and 10M steps), and sampling-step counts. The only difference is the model formulation itself, as we state in Section 6, to keep the comparison as fair as possible.

---

### Review · Reviewer_Suqn · 2026-05-31

**Summary Of Contributions:**

The paper proposes the use of Bayesian belief updates to characterize a reverse process akin to the diffusion forward process in which samples are recursively generated as noise realizations of the target with more and more confidence exploring that the Bayesian update rule naturally reduces variance as samples are successively generated and the posterior distribution updated. The approach starts from a wide distribution and converges upon having recursively generated sufficient numbers of samples to a sufficiently reduced posterior variance. The procedure relies on a denoising network what produces the updated posterior mean when given current mean and variance as input at each successive step of the generative process, i.e. reflecting the denoising network in a diffusion model. An ELBO is derived where the q distributions reflect the updated beliefs and the ELBO controlled using a  fixed noise variance scheduler. The proposed approach is compared to Bayesian Flow Networks (BFNs) as well as Variational Diffusion Models (VDM) providing samples of high quality based on a simple to implement algorithmic framework. Theoretical properties are further carefully described and derived as well as the relation of the proposed approach to BFN and VDM.

**Strengths**
*  The paper is well written and easy to follow with the methodology well backed up by theory.
*  The approach is elegant in exploring Bayesian update of beliefs as a simple reverse process in the context of diffusion with the forward model in the supplementary nicely demonstrated to correspond to a non-Markovian diffusion forward process.
* The experimentation highlights the utility of the method.
* The approach is well related to BFN and VDM and appears easy to implement also producing samples of good quality in terms of the FID metric when compared to BFN and VDM.

**Weaknesses:**
* The comparison is reduced to BFN and VDM. While this makes sense as the approach can be directly related to these approaches it would be interesting to also relate and discuss the proposed approach in terms of the wider use of sequential Monte-Carlo and importance sampling based procedures in diffusion models (see also requested changes).
* The experimentation is executed on limited but sufficient amount of data, i.e. ImageNet and Cifar10. Whereas FID gains are observed there are not any gains in terms of BPD. It would be good to elaborate upon why gains are not consistent across the two metrics.
* The approach relies on the precision scheduler \alpha_i. It would be good to include analysis of the influence of this scheduler as well as a discussion on the design choices of this scheduler – also it would be interesting to relate this scheduler to the corresponding diffusion steps and how the steps can be kept at a minimum.

**Audience:**

Yes

**Audience Explanation:**

The proposed method is highly relevant to the TMLR community and provides an interesting and simple to implement generative framework relevant also specifically in the context of diffusion modeling approaches. The approach here provides an interesting contribution to the community.

**Broader Impact Concerns:**

There are no immediate broader impact concerns, however, the manuscript could potentially discuss general concerns in terms of the broader impact of generative modeling in general including issues of the creation of deep fakes etc.

**Claims And Evidence:**

Yes

**Claims Explanation:**

The claims are generally well supported and the experimentation reasonable and well compared and related to BFN and VDM.
It would be good to include analyses of choices of schedulers for \alpha_i and also explicitly demonstrate that L_R has minimal influence and can be ignored when training as stated. It seems this would depend on the choice of \alpha_R.

**Requested Changes:**

Overall I like this paper and find it well written and executed providing also good theoretical support for the developed methodology and suitable for publication at TMLR. The requested adjustment are therefore not to be treated as critical but as constructive suggestions for improvements.

Whereas the positioning of the paper with respect to BFN and VDM is clear and well described there are sequential Monte-Carlo approaches also utilizing importance sampling recently developed that also directly explores the reverse process in a Bayesian inference context. This includes:
https://proceedings.neurips.cc/paper_files/paper/2025/file/6490ab6ef8eb7116c6c1889795d21748-Paper-Conference.pdf

as well as  ”Sequential Controlled Langevin Diffusions” exploring sequential Monte-Carlo and diffusion available at
https://arxiv.org/html/2412.07081v1

It would thus be interesting to also discuss these sequential MC procedures in the context of the presented work. The authors do not need to explicitly compare to these procedures but at least including these procedures as context would strengthen the paper.

Why are gains only observed for FID and not BPD – this would be good to further elaborate upon?

Can you further clarify why L_R can be ignored in the training – this as I see it must also depend on the choice of \alpha_R. Would be good to demonstrate the effect of it being diminishing and ignorable in the supplementary to support this.

It would further be good to elaborate on the choices of variance schedulers for \alpha_i and the influence upon the performance of choices of this scheduler. This as I see it would potentially also influence the number of required steps etc.

---

> ### Author Response · Authors · 2026-06-11
> **Rebuttal**
>
> Thank you for your review and for recognizing the theoretical grounding and elegance of our approach.
>
> **SMC**: Thank you for pointing us to these interesting works. We will add a further section to Appendix A to describe how BSI relates to these approaches. In general, these methods solve a somewhat different problem (sampling from an unnormalized density) compared to BSI (a generative model derived from a posterior inference perspective).
>
> **FID vs. BPD**: The log-likelihoods are similar because the three models have an ELBO of the form in Equation (14): an integral of the expected reconstruction error across noise levels. While the encoding distributions are different, they are similar enough that the decoder achieves comparable reconstruction errors and thus comparable ELBOs. However, the FID differs, because here we take discrete steps through the noise level range and thus the noise level through time differs a lot. Figure 2 shows that BFNs, for example, have little noise and only for a brief period in the sampling process, which means that only a few steps actually contribute the high-level features of the sample. In contrast, BSI has more noise over a wider range and thus produces better samples with fewer steps as we have shown in our experiments.
>
> **$L_R$**: The reconstruction term is negligible because of the total accumulated precision, not the reconstruction precision. By the time we have accumulated $\alpha_M$ precision, the belief is so concentrated that essentially all of its remaining mass about each pixel falls within a single 8-bit color bucket of width 1/256. The final reconstruction is therefore all but determined and contributes negligibly to the discretized likelihood. This reasoning is also in Section 5.3.
>
> **Precision schedule**: For this work, we chose to focus on the simple uniform step schedule against the ELBO encoder / training distribution. While other schedules are easily possible, we deliberate decided not to tune this aspect to keep the comparison to VDMs and BFNs fair and direct, since the baselines also do not tune their schedules specifically for sample quality.
>
> We will add a broader impact paragraph on the potential risks of generative models.